# Ripple Perturbations Through Structure: Likelihood-Constrained Adversarial Attacks on Heterogeneous Tabular Data

**Zhengjie Zhou** [1]  **Jiahuan Yan** [1]  **Boqun Ma** [1]  **Weiwei Feng** [1]  **Tengfei Liu** [1]  **Weiqiang Wang** [1]

## Abstract

Generating realistic adversarial examples for tabular data remains challenging due to heterogeneous feature types and asymmetric inter-feature dependencies. Existing approaches typically rely on handcrafted constraints or undirected similarity criteria to delimit the feasible attack region, which often fail to capture the structural dependency governing tabular generation. Consequently, standard attacks typically produce perturbations that are statistically likely yet semantically inconsistent and prone to optimization stagnation via gradient masking. To address this, we propose LCSA, a white-box framework that formulates adversarial generation as optimization over structurally admissible perturbations. LCSA leverages an ensemble of heterogeneous neural Structural Causal Models to infer dependencies and introduces a structure-aware ripple mechanism. Unlike attacks that perturb features in isolation, this mechanism propagates updates downstream, acting as a structural preconditioner that conditions gradient flow to mitigate masking effects. Extensive experiments demonstrate that LCSA outperforms state-of-the-art baselines in 45 of 50 evaluated configurations, yielding adversarial examples with superior structural consistency and transferability.

## 1. Introduction

Adversarial attacks seek to introduce imperceptible perturbations into input data to mislead predictive models. While such attacks have been extensively explored in computer vision and natural language processing (Ma et al., 2021; Wang et al., 2023), the vulnerability of deep tabular models remains substantially underexplored. This gap is critical, given the increasing deployment of deep tabular architectures (Chen et al., 2024; Yan et al., 2023) in high-stakes domains such as finance, healthcare, and fraud detection (Borisov et al., 2022). Unlike homogeneous image data where slight pixel shifts are visually imperceptible, tabular features often represent discrete entities governed by strict logic (e.g., age is an integer, birth date before death). Consequently, imperceptibility in this domain demands more than geometric proximity; it requires adherence to the *intrinsic dependency structure and logical coherence*, ensuring that perturbations respect the rigid constraints.

Current approaches to tabular adversarial attacks struggle to model plausibility beyond statistical validity. While early methods relied on handcrafted constraints to delimit the feasible attack regions (Ballet et al., 2019; Gressel et al., 2021; Simonetto et al., 2024a;b), recent automated frameworks have sought to implicitly define these regions via *dependency mining* (Ben-Tov et al., 2024; Bostani et al., 2022; Mathov et al., 2022; Sheatsley et al., 2020) or *generative modeling* (Cartella et al., 2021; He et al., 2025c; Kotelnikov et al., 2023). However, as illustrated in the top panel of Fig. 1, dependency mining is restricted to a sparse subset of high-confidence rules, failing to capture the dense web of global interactions. Conversely, generative approaches, shown in the middle panel of Fig. 1, implicitly define feasible regions but rely on **undirected associative correlations**. By modeling the data manifold as isotropic or symmetric, both paradigms fail to capture the asymmetric dependencies inherent to tabular data (e.g., *income → tax bracket*). Consequently, these methods produce perturbations that are statistically likely yet semantically inconsistent, violating the coherence required for valid tabular samples.

To bridge the gap between statistical validity and structural plausibility, we propose the **Likelihood-Constrained Structural Attack (LCSA)**. Unlike standard methods that perturb features in isolation, LCSA formulates adversarial generation as a structure-aware optimization process over structurally admissible perturbations. The framework operates by first learning an ensemble of heterogeneous neural structural causal models (SCM) to encode the approximate directed dependencies. Subsequently, we introduce a **ripple perturbation mechanism** that fundamentally *reparameter-*

[1] Ant Group, Hangzhou, China. Correspondence to: Tengfei Liu <aaron.ltf@antgroup.com>, Zhengjie Zhou <zhouzhengjie.zzj@antgroup.com>.

*Proceedings of the $43^{rd}$ International Conference on Machine Learning*, Seoul, South Korea. PMLR 306, 2026. Copyright 2026 by the author(s).

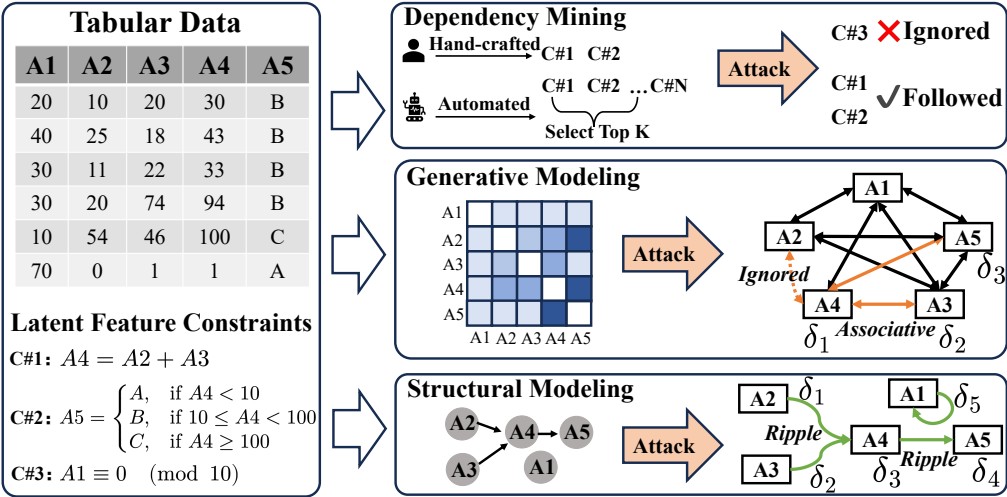

*Figure 1.* Illustration of structural constraints in tabular attacks. Unlike Dependency Mining which misses global constraints (top), or Generative Modeling (e.g., Variational Autoencoders) which optimizes over undirected associations (middle), our LCSA framework (bottom) models directed dependencies. This enforces logical validity by ensuring that adversarial perturbations propagate coherently (visualized as Ripple effects) and respect intrinsic node constraints (e.g., constraints on $A1$).

*izes* the adversarial optimization. LCSA forces perturbations to propagate downstream via the learned SCMs. Functionally, this mechanism resolves the tension between constraint satisfaction and attack efficacy: it conditions the gradient flow along the data manifold. This facilitates the optimizer in escaping local optima and gradient masking common in heterogeneous spaces, strictly adhering to both geometric budgets and the structural constraints of mixed-type data.

Our contributions are summarized as follows:

**(1)** We identify a fundamental blind spot in existing tabular attacks: the reliance on symmetric optimization that ignores the **consistency of feature interactions**. We redefine adversarial imperceptibility by enforcing **structural admissibility**, constraining the search space to perturbations that respect the logical coherence of the data generating process.

**(2)** We propose LCSA, a unified framework that embeds **Heterogeneous Neural Structural Causal Models** into the adversarial loop. Via a novel **structure-aware ripple mechanism**, LCSA models perturbations as dynamic interventions that induce cascading updates. We provide a theoretical analysis showing that this mechanism aggregates downstream gradients to mitigate masking effects, facilitating optimization in regions where other baselines stagnate.

**(3)** Extensive experiments on five real-world and three synthetic datasets demonstrate that LCSA outperforms state-of-the-art baselines in **45 out of 50 experimental settings**. Crucially, our synthetic analysis provides the rigorous quantification of structural consistency, demonstrating robustness even under SCM estimation errors.

**Conflict of Interest Disclosure:** All authors are employed by Ant Group. Since this work does not evaluate or promote any proprietary models or products developed by our employer, we have no conflicts of interest to disclose.

## 2. Preliminaries

In this section, we formalize the adversarial attack setting for tabular data and introduce the factorized structural representation. Our goal is to demonstrate how the generative structure of SCMs defines a likelihood-based feasible region. This region complements traditional geometric constraints by enforcing *logical coherence* in heterogeneous domains.

**Adversarial Attacks on Tabular Data.** Let $\mathcal{D}_{data} = \{(x, y)\}$ denote a tabular dataset, where $x$ represents a feature vector and $y \in \{1, \dots, C\}$ is the class label. Unlike homogeneous image data, tabular features are typically *heterogeneous*. We denote the sets of continuous and categorical feature indices as $\mathcal{I}_{\text{cont}}$ and $\mathcal{I}_{\text{cat}}$, respectively. Consequently, the feature space is defined as $\mathcal{X} = \mathbb{R}^{|\mathcal{I}_{\text{cont}}|} \times \prod_{j \in \mathcal{I}_{\text{cat}}} \Delta^{K_j - 1}$, where $|\mathcal{I}_{\text{cont}}|$ is the number of continuous features and $\Delta^{K_j - 1}$ represents the probability simplex for a categorical feature with $K_j$ categories. Each sample $x = (x_1, x_2, \dots, x_d) \in \mathcal{X}$ consists of $d$ variables, where each variable $x_j$ is either a continuous feature taking values in $\mathbb{R}$ for $j \in \mathcal{I}_{\text{cont}}$, or a categorical feature represented as a probability vector on the simplex $\Delta^{K_j - 1}$ for $j \in \mathcal{I}_{\text{cat}}$.

Given a classifier $h_\theta : \mathcal{X} \to \mathbb{R}^C$, a standard adversarial attack seeks a perturbation $\delta$ such that the perturbed instance $\tilde{x} = x + \delta$ induces misclassification while remaining imperceptibly close to $x$. This is commonly formulated as a constrained optimization problem:

$$\max_{\delta} \mathcal{L}_{cls}\big(h_\theta(x + \delta), y\big) \quad \text{s.t.} \quad \|\delta\|_p \leq \epsilon, \qquad (1)$$

where $\mathcal{L}_{cls}$ is the classification loss (i.e., cross-entropy loss) and $\epsilon$ is the geometric perturbation budget under an $\ell_p$ norm.

While this formulation ensures geometric proximity, it fails to account for the complex structural dependencies inherent to tabular data. Note that for $j \in \mathcal{I}_{\text{cat}}$, $x_j + \delta_j \in \Delta^{K_j - 1}$.

**Heterogeneous Structural Causal Models.** As a widely adopted paradigm for modeling structural dependencies in the tabular domain (Deleu et al., 2022; Löwe et al., 2022), we define the Structural Causal Model, denoted as $\mathcal{M} = (\mathcal{G}, F, P_Z)$. Here, $\mathcal{G} = (\mathcal{V}, \mathcal{E})$ is a Directed Acyclic Graph (DAG) with nodes $\mathcal{V} = \{1, \ldots, d\}$ representing features and edges $\mathcal{E}$ encoding parent-child dependencies. The symbol $F = \{f_1, \ldots, f_d\}$ represents the set of deterministic structural functions, and $P_Z$ denotes the joint distribution of independent exogenous noise variables $Z = \{Z_1, \ldots, Z_d\}$. Each feature $x_j$ is generated by a structural assignment:

$$x_j = f_j(x_{\text{PA}_j}, Z_j), \qquad j = 1, \ldots, d, \qquad (2)$$

where $x_{\text{PA}_j}$ denotes the set of parent variables of $x_j$ in the graph $\mathcal{G}$, and $Z_j$ is an exogenous noise variable drawn from the marginal distribution $P_Z$. The SCM factorizes the joint distribution $p_{\text{SCM}}(x)$ as:

$$p_{\text{SCM}}(x) = \prod_{j=1}^{d} p(x_j \mid x_{\text{PA}_j}). \qquad (3)$$

The negative log-likelihood of a sample $x$, which serves as a surrogate measure of structural violation under the assumed model (Bishop, 1993; Song et al., 2018), is given by:

$$-\log p_{\text{SCM}}(x) = \sum_{j=1}^{d} -\log p(x_j \mid x_{\text{PA}_j}). \qquad (4)$$

This formulation provides a probabilistic definition of feasibility that applies to both continuous and discrete modalities.

**Likelihood-Constrained Tabular Attack.** We characterize valid perturbations as those residing in high-density regions of a *SCM-induced generative model learned from observational data*. We define the *Structural Violation Score* of a perturbed sample $\tilde{x} = x + \delta$ as its negative log-likelihood: $\mathcal{L}_{\text{scm}}(\tilde{x}) = -\log p_{\text{SCM}}(\tilde{x})$. Accordingly, we define the *Structural Feasible Set* $\Delta_{\text{scm}}$ as the set of perturbations whose induced likelihood deviation remains within the learned structural constraint:

$$\Delta_{\text{scm}} = \{\delta \mid \mathcal{L}_{\text{scm}}(\tilde{x}) - \mathcal{L}_{\text{scm}}(x) \leq \gamma\}, \qquad (5)$$

where $\gamma$ is a likelihood-based structural budget controlling the allowable deviation under $p_{\text{SCM}}$. Consequently, the likelihood-constrained tabular attack is formulated as:

$$\max_{\delta} \mathcal{L}_{cls}(h_\theta(x + \delta), y) \quad \text{s.t.} \quad \delta \in \Delta_{geo} \cap \Delta_{\text{scm}}, \qquad (6)$$

where $\Delta_{geo} = \{\delta \mid \|\delta\|_p \leq \epsilon\}$ represents the geometric constraint. This unified formulation ensures that the perturbation remains both geometrically imperceptible and structurally consistent under the assumed generative model.

## 3. Methodology

In this section, we present the **Likelihood-Constrained Structural Attack (LCSA)**, a white-box framework that redefines adversarial generation by strictly enforcing *structural plausibility* alongside geometric imperceptibility. LCSA operates through a synergistic two-phase framework: **Phase I: Heterogeneous Ensemble SCM Learning** and **Phase II: Structure-Aware Adversarial Optimization**. Phase I constructs a probabilistic foundation by inferring an ensemble of heterogeneous SCMs from observational data. Building upon this learned structure, Phase II formulates the attack as a constrained optimization problem.

### 3.1. Phase I: Heterogeneous Ensemble SCM Learning

Standard SCMs typically assume homogeneous Gaussian noise (Johansson, 2024), which is incompatible with the mixed feature space $\mathcal{X}$. To address this, we construct an ensemble of **Heterogeneous Neural SCMs** that model distinct mechanisms for continuous and categorical data.

**Mixed-Type Structural Mechanisms.** We model the conditional mechanism of each feature $x_j$. To mitigate finite-sample noise and capture epistemic uncertainty, we train an ensemble of $M$ models on bootstrap subsamples. For the $m$-th model, the prediction $\hat{x}_j^{(m)}$ is derived from its parents $x_{\text{PA}_j}^{(m)}$ according to the variable type:

$$\hat{x}_j^{(m)} = \begin{cases} \text{MLP}_j^{(m)}(x_{\text{PA}_j}^{(m)}; \phi_j^{(m)}) & \text{if } j \in \mathcal{I}_{\text{cont}}, \\ \text{Softmax}\left(\text{MLP}_j^{(m)}(x_{\text{PA}_j}^{(m)}; \phi_j^{(m)})\right) & \text{if } j \in \mathcal{I}_{\text{cat}}, \end{cases} \qquad (7)$$

where $\text{MLP}_j^{(m)}$ is a multilayer perceptron parameterized by weights $\phi_j^{(m)}$. For continuous features $j \in \mathcal{I}_{\text{cont}}$, the output estimates the value directly. For categorical features $j \in \mathcal{I}_{\text{cat}}$, the output is a probability vector in the simplex $\Delta^{K_j - 1}$. The set of all functional parameters for the $m$-th model is denoted by $\Phi^{(m)} = \{\phi_j^{(m)}\}_{j=1}^{d}$ and $\hat{x}^{(m)} = \{\hat{x}_1^{(m)}, \ldots, \hat{x}_d^{(m)}\}$. To strictly enforce the structural dependencies within the MLPs, we apply a block-wise mask to their input layers. This mask handles the dimension mismatch caused by one-hot encoding and ensures that each function depends solely on its valid parents.

**Differentiable Ensemble Structure Learning.** Since the parent sets are initially unknown, we explicitly parameterize the structure using a learnable weighted adjacency matrix $W^{(m)} \in \mathbb{R}^{d \times d}$ for each model. The block-wise mask described above is dynamically projected from this matrix, where non-zero entries $W_{i,j}^{(m)}$ activate the corresponding feature blocks in the input layer of $\text{MLP}_j^{(m)}$, effectively treating $x_i$ as a component of the parent vector (i.e., $x_i \in x_{\text{PA}_j}^{(m)}$). We jointly optimize the weighted adjacency matrices $W^{(m)}$ and the functional parameters $\Phi^{(m)}$ by minimizing a het-

erogeneous reconstruction error subject to the acyclicity constraint. The optimization objective is formulated as:

$$\min_{\Phi, W} \sum_{m=1}^{M} \sum_{x \in \mathcal{D}_{data}} \left[ \mathcal{L}_{\text{recon}}(x, \hat{x}^{(m)}) + \|W^{(m)}\|_1 + \|W^{(m)}\|_{\text{tr}} \right],$$
(8)

where the heterogeneous reconstruction loss $\mathcal{L}_{\text{recon}}$ combines Mean Squared Error for continuous variables and Cross-Entropy for categorical variables:

$$\mathcal{L}_{\text{recon}}(x, \hat{x}^{(m)}) = \sum_{j \in \mathcal{I}_{\text{cont}}} \|x_j - \hat{x}_j^{(m)}\|^2 + \sum_{j \in \mathcal{I}_{\text{cat}}} \text{CE}(x_j, \hat{x}_j^{(m)}).$$
(9)

The term $\|W^{(m)}\|_{\text{tr}}$ enforces the DAG constraint via the log-determinant characterization (Bello et al., 2022):

$$\|W^{(m)}\|_{\text{tr}} = -\log \det(I - W^{(m)} \odot W^{(m)}),$$
(10)

where $\det(\cdot)$ denotes the matrix determinant, $I \in \mathbb{R}^{d \times d}$ is the identity matrix, and $\odot$ represents the Hadamard product. For high-dimensional datasets, $W^{(m)}$ is further masked by a sparse candidate set to ensure computational feasibility. See **Appendix E** for more implementation details.

**Exogenous Noise Estimation.** A critical component of our likelihood constraint is the estimation of residual uncertainty under the learned mechanisms. After training, we compute the residuals for each feature $j \in \mathcal{I}_{\text{cont}}$ as $R_j^{(m)} = \{x_j - \hat{x}_j^{(m)} \mid x \in \mathcal{D}_{data}\}$. We then estimate the variance:

$$\hat{\sigma}_{j,m}^2 = \text{Var}(R_j^{(m)}).$$
(11)

These variance terms serve as normalization factors in the subsequent attack phase, ensuring the perturbation budget adapts to the intrinsic stochasticity of each feature. Note that for categorical features $j \in \mathcal{I}_{\text{cat}}$, explicit variance estimation is unnecessary, which naturally scales the likelihood contribution via Cross Entropy (Nazabal et al., 2020).

### 3.2. Phase II: Structure-Aware Adversarial Optimization

Given the learned ensemble, we optimize the structured perturbation to maximize the classification loss under likelihood-based structural and geometric constraints.

**Feasible Region and Simplex Constraints.** Since tabular features are heterogeneous, the perturbation $\delta = \{\delta_1, \delta_2, \ldots, \delta_d\}$ must satisfy distinct constraints. For continuous features, we bound the magnitude via the $\ell_p$-norm. For a categorical feature $j$, the perturbed vector must reside on the simplex $\Delta^{K_j - 1}$ during adversarial optimization. Thus, the feasible set $\Omega$ is defined as:

$$\Omega = \left\{ \delta \mid \|\delta\|_p \leq \epsilon, \quad \forall j \in \mathcal{I}_{\text{cat}} : (x_j + \delta_j) \in \Delta^{K_j - 1} \right\}.$$
(12)

**Soft-Embedding Relaxation.** Standard tabular models utilize discrete index lookups for categorical features (Arik &

Pfister, 2021). This operation is non-differentiable. To enable optimization, we replace the discrete lookup with a differentiable **Soft-Embedding** operation. Let $\mathbf{E}_j \in \mathbb{R}^{K_j \times d_{\text{emb}}}$ be the embedding matrix of the target model $h_\theta$, with $d_{\text{emb}}$ denoting the dimension of the dense vector space. We compute the continuous input representation $z_j$ as the expectation of the embeddings weighted by $x_j$ and $\delta \in \Omega$:

$$z_j = (x_j + \delta_j)^\top \mathbf{E}_j, \quad j \in \mathcal{I}_{\text{cat}}.$$
(13)

By feeding this continuous $z_j$ into the target model $h_\theta$, we establish a fully differentiable path from the classification loss back to the perturbation $\delta$.

**Structure-Aware Ripple Perturbation.** Instead of perturbing features in isolation, we model a *ripple effect* where perturbations on parent variables propagate to descendants through the learned directed dependencies. We define the adversarial example recursively based on the topological order. For each feature $x_j$, the perturbation is:

$$\delta_j = \underbrace{\nu_j}_{\text{Latent Update}} + \underbrace{\frac{1}{M} \sum_{m=1}^{M} \left( \hat{f}_j^{(m)}(\tilde{x}_{\text{PA}_j}^{(m)}) - \hat{f}_j^{(m)}(x_{\text{PA}_j}^{(m)}) \right)}_{\text{Structural Ripple}},$$
(14)

where $\hat{f}_j^{(m)}(\cdot)$ is the ensemble mechanism learned in Eq. 7, $\nu_j$ is the latent update variable, and $\tilde{x}_{\text{PA}_j}^{(m)} = x_{\text{PA}_j}^{(m)} + \delta_{\text{PA}_j}$. This mechanism ensures that any update to a parent feature triggers a consistent adjustment in its children, maintaining logical coherence. Functionally, Eq. (14) serves as a structural reparameterization that conditions the gradient flow (see **Appendix D** for a theoretical derivation). To bridge the discrete-continuous gap, we relax categorical parents in $\tilde{x}_{\text{PA}_j}^{(m)}$ to probability vectors on the simplex $\Delta^{K-1}$. This formulation treats the SCM's input layer as a linear embedding lookup, allowing the perturbed structure to be optimized via valid semantic interpolation of the learned feature space.

**Ensemble Structural Violation.** With the recursively constructed $\tilde{x} = \{\tilde{x}_1, \tilde{x}_2, \ldots, \tilde{x}_d\}$ and $\tilde{x}_d = x_d + \delta_d$, we quantify structural deviation using a differentiable surrogate of the SCM negative log-likelihood under the learned ensemble. We define the structural loss $\mathcal{L}_{\text{scm}}$ as:

$$\mathcal{L}_{\text{scm}}(\tilde{x}) = \frac{1}{M} \sum_{m=1}^{M} \left[ \sum_{j \in \mathcal{I}_{\text{cont}}} \frac{\|\tilde{x}_j - \hat{f}_j^{(m)}(\tilde{x}_{\text{PA}_j}^{(m)})\|^2}{2\hat{\sigma}_{j,m}^2} + \sum_{j \in \mathcal{I}_{\text{cat}}} -\tilde{x}_j^\top \log \hat{f}_j^{(m)}(\tilde{x}_{\text{PA}_j}^{(m)}) \right].$$
(15)

**Dual-Ascent Optimization.** To improve convergence stability, we employ the **Augmented Lagrangian Method** (Bertsekas, 2014; Nocedal & Wright, 2006). We formulate the optimization over the latent update variable $\nu$, regarding the perturbation $\delta$ as a function of $\nu$ dictated by Eq. (14).

We adopt a two-stage strategy consisting of a *Warm Start* and a *Constraint Correction*.

*Initialization.* We initialize the update vector $\nu^{(0)} = \mathbf{0}$ and the Lagrange multiplier $\lambda^{(0)} = 0$. Let $\eta > 0$ denote the step size for the gradient ascent and $\rho > 0$ be the penalty parameter controlling the quadratic regularization strength.

*Stage 1: Warm Start.* In the first $T_{\text{warm}}$ iterations, we optimize solely for the classification objective. For $t = 0, \ldots, T_{\text{warm}} - 1$, we update $\nu$ via projected gradient ascent:

$$\nu^{(t+1)} = \nu^{(t)} + \eta \nabla_\nu \mathcal{L}_{\text{cls}}(h_\theta(x + \Pi_\Omega(\delta(\nu^{(t)}))), y), \quad (16)$$

where $\delta(\nu^{(t)})$ denotes the total perturbation vector computed recursively via Eq. (14) given the current $\nu^{(t)}$ and $\mathcal{L}_{\text{cls}}$ is the classification loss in Eq. 1. The operator $\Pi_\Omega$ denotes the projection operator that enforces feasibility. For continuous features $j \in \mathcal{I}_{\text{cont}}$, we project $\delta_j$ onto the $\ell_p$-norm ball (via clipping for $L_\infty$ or rescaling $\delta_j \cdot \min(1, \frac{\epsilon}{\|\delta_j\|_2})$ for $L_2$). For categorical features $j \in \mathcal{I}_{\text{cat}}$, we apply the Euclidean projection onto the probability simplex (Duchi et al., 2008).

*Stage 2: Constraint Correction.* After the warm start, we activate the likelihood-based structural constraint according to $\mathcal{L}_{\text{scm}}$ (Eq. 15). For iterations $t = T_{\text{warm}}, \ldots, T_{\text{total}}$, we update $\nu$ to maximize the augmented Lagrangian objective:

$$\nu^{(t+1)} = \nu^{(t)} + \eta \nabla_\nu \Big[ \mathcal{L}_{\text{cls}}(h_\theta(\tilde{x}^{(t)}), y) \\ - \lambda^{(t)} \psi(\tilde{x}^{(t)}) - \frac{\rho}{2} \max(0, \psi(\tilde{x}^{(t)}))^2 \Big], \quad (17)$$

where $\tilde{x}^{(t)} = x + \Pi_\Omega(\delta(\nu^{(t)}))$ is the adversarial example at step $t$, and $\psi(\tilde{x}^{(t)}) = \mathcal{L}_{\text{scm}}(\tilde{x}^{(t)}) - \mathcal{L}_{\text{scm}}(x) - \gamma$ represents the constraint violation measure. The gradients $\nabla_\nu$ are computed via backpropagation through the recursive ripple structure. Simultaneously, the Lagrange multiplier $\lambda$ is updated based on the violation:

$$\lambda^{(t+1)} = \max\left(0, \lambda^{(t)} + \rho \cdot \psi(\tilde{x}^{(t+1)})\right). \quad (18)$$

This adaptive update increases $\lambda$ to the persistence of the violation $\psi(\cdot)$, encouraging the final $\nu$ to generate perturbations that satisfy the structural constraint $\Delta_{\text{scm}}$.

**Final Optimization Objective.** We explicitly formulate LCSA as the following constrained maximization problem:

$$\max_\nu \quad \mathcal{L}_{\text{cls}}\big(h_\theta(\tilde{x}^{(t)}), y\big) \quad \text{s.t.} \quad \begin{cases} \mathcal{L}_{\text{scm}}(\tilde{x}^{(t)}) - \mathcal{L}_{\text{scm}}(x) \leq \gamma, \\ \delta \in \Omega, \end{cases} \quad (19)$$

where $\delta$ is implicitly defined as a deterministic function of $\nu$ via the ripple reparameterization in Eq. (14). The first constraint enforces adherence to high-density regions under the learned heterogeneous SCM ensemble. The second constraint $\Omega$ ensures geometric imperceptibility and mathematical validity on the probability simplex. Upon convergence or after $T_{\text{total}}$ iterations, $x_{adv} = \tilde{x}^{(T_{\text{total}})}$.

*Table 1.* Specification of Synthetic Structural Benchmarks. $\mathcal{U}$ denotes the Uniform noise and $\mathcal{N}$ represents Gaussian noise, and $\mathbb{I}(\cdot)$ is an indicator function

| Topology | Structural Mechanism | Target Label |
|---|---|---|
| **Chain** $X_1 \to X_2 \to X_3 \to X_4$ | $x_1 \sim \mathcal{U}(0, 2)$ $x_2 = 2x_1 + \mathcal{N}(0, 0.01)$ $x_3 = x_2 + 0.5x_1 + \mathcal{N}(0, 0.01)$ $x_4 = \tanh(x_3) + \mathcal{N}(0, 0.01)$ | $y = \mathbb{I}(x_3 + \log(x_2 + 1) \leq 1)$ |
| **Collider** $X_1 \to X_3 \leftarrow X_2$ $X_3 \to X_4$ | $x_1, x_2 \sim \mathcal{N}(0, 1)$ $x_3 = x_1 + x_2 + \mathcal{N}(0, 0.01)$ $x_4 = \text{Sigmoid}(x_3) + \mathcal{N}(0, 0.01)$ | $y = \mathbb{I}(x_3 - x_4 \leq 0.2)$ |
| **Confounder** $X_2 \leftarrow X_1 \to X_3$ $X_3 \to X_4$ | $x_1 \sim \mathcal{U}(-1, 1)$ $x_2 = 3x_1 + \mathcal{N}(0, 0.01)$ $x_3 = e^{x_1} + \mathcal{N}(0, 0.01)$ $x_4 = 2x_3 + 1 + \mathcal{N}(0, 0.01)$ | $y = \mathbb{I}(x_3 \cdot e^{x_4} \leq e)$ |
| **Cyclic** $X_1 \to X_2 \to X_3 \to X_4 \to X_2$ | $x_1 \sim \mathcal{U}(-1, 1)$ $x_2 = 2x_1 + 0.3x_4 + \mathcal{N}(0, 0.01)$ $x_3 = \tanh(x_2) \times (1 + \mathcal{N}(0, 0.01))$ $x_4 = 0.5x_3 \times (1 + \mathcal{N}(0, 0.01))$ | $y = \mathbb{I}(x_3 \cdot x_4 \leq 1)$ |

**Post-Optimization Verification.** The structural constraint is enforced through the augmented penalty during optimization. Since categorical variables are optimized in a relaxed simplex space, we discretize them after convergence by selecting the category with maximum probability. We finally retain only adversarial examples that both fool the classifier and satisfy the structural constraint $\Delta_{\text{scm}}$.

The detailed algorithm, complexity analysis, and theoretical justification are provided in **Appendix B**, **C**, and **D**.

## 4. Experiments

In this section, we conduct experiments on five real-world and three synthetic datasets to evaluate the performance of LCSA, focusing on the following research questions:

- **RQ1 (Attack Effectiveness):** Does LCSA achieve competitive robust accuracy against state-of-the-art methods, particularly under complex constraint settings?
- **RQ2 (Tolerance):** How resilient is LCSA to tightening geometric ($\epsilon$) and structural ($\gamma$) budgets?
- **RQ3 (Robustness):** How does estimation noise within the learned SCMs impact the attack's overall performance?
- **RQ4 (Computational Efficiency):** What is the computational overhead of the LCSA framework?
- **RQ5 (Scalability Study):** How is LCSA scalable, especially in terms of transferability and adversarial training?
- **RQ6 (Practical Stealthiness):** Can LCSA evade anomaly detection pipelines in deployment scenarios?
- **RQ7 (Ablation Study & Hyperparameter Sensitivity):** How do key components and hyperparameter settings contribute to the attack's stability and efficacy?

### 4.1. Experimental settings

**Real-World Dataset.** Following previous works (Simonetto et al., 2024a;b;c), We evaluate on five real-world datasets: URL (Hannousse & Yahiouche, 2021), LCLD (George, 2018), WiDS (Lee et al., 2020), CTU (Chernikova & Oprea, 2022) and Malware (Dyrmishi et al., 2023).

**Synthetic Structural Dataset.** To address **RQ1** and **RQ3**, we construct four synthetic benchmarks representing fundamental structural topologies: Chain, Collider, Confounder

*Table 2.* Evaluation on robust accuracy. We compare LCSA against gradient-based (PGD, CaFA, CAPGD), generative-manifold (Adv-VAE), and heuristic (BF*, MOEVA) baselines under both $L_{\{2,\infty\}}$ and $\gamma$ constraints. The Clean column reports accuracy on unperturbed samples; lower robust accuracy indicates a more effective attack. We highlight the lowest robust accuracy in **bold**.

| Dataset | Model | Clean | $L_2 = 0.5 \& \gamma = 0.5$ | | | | | | | $L_\infty = 0.5 \& \gamma = 0.5$ | | | | | | |
| --- | --- | --- | PGD | Adv-VAE | CaFA | CAPGD | BF* | MOEVA | LCSA | PGD | Adv-VAE | CaFA | CAPGD | BF* | MOEVA | LCSA |
| URL | TabTr. | 93.6 | 34.7±0.9 | 28.4±1.3 | 17.2±1.1 | 11.3±0.4 | 92.8±0.2 | 54.1±1.8 | **2.1±0.3** | 38.2±1.2 | 31.5±1.0 | 27.5±0.9 | 20.8±0.5 | 93.1±0.3 | 59.4±2.1 | **3.0±0.7** |
| | RLN | 94.4 | 37.1±0.8 | 29.6±0.4 | 20.1±0.5 | 12.9±0.3 | 94.1±0.1 | 61.5±1.3 | **6.7±0.5** | 42.3±1.1 | 35.8±1.2 | 30.4±0.7 | 23.1±0.6 | 93.9±0.5 | 66.2±1.5 | **9.4±0.8** |
| | VIME | 92.5 | 64.2±1.0 | 60.1±1.2 | 58.7±0.2 | 56.4±0.5 | 91.8±0.2 | 79.3±1.9 | **19.6±0.9** | 67.5±1.2 | 61.2±1.4 | 44.1±0.8 | 38.9±0.6 | 92.0±0.4 | 77.5±2.2 | **21.3±0.7** |
| | STG | 93.3 | 80.1±0.7 | 76.5±0.8 | 74.5±0.4 | 73.1±0.3 | 92.9±0.1 | 84.7±1.1 | **14.8±0.4** | 47.2±0.9 | 40.1±0.8 | 31.4±0.6 | 27.5±0.4 | 93.0±0.1 | 54.9±1.6 | **16.9±0.6** |
| | TabNet | 93.4 | 47.5±0.9 | 40.2±1.0 | 28.1±0.6 | 20.5±0.5 | 91.2±0.2 | 67.8±1.4 | **6.8±0.3** | 50.6±1.0 | 44.5±1.1 | 45.3±0.8 | 35.8±0.6 | 91.7±0.7 | 69.4±1.8 | **7.5±0.4** |
| LCLD | TabTr. | 69.5 | 45.8±1.4 | 38.2±1.3 | 33.7±0.9 | 27.9±0.6 | 61.8±0.3 | 57.2±1.7 | **5.1±0.4** | 23.1±1.6 | 18.5±1.4 | 13.9±0.8 | 11.2±0.5 | 60.1±0.4 | 26.8±2.0 | **7.8±0.6** |
| | RLN | 68.3 | 17.5±1.0 | 12.6±0.9 | 6.9±0.6 | 0.9±0.2 | 39.5±0.6 | 24.9±1.3 | **0.2±0.1** | 11.4±1.1 | 9.2±0.8 | 8.8±0.7 | 6.5±0.5 | 37.8±0.7 | 14.7±1.6 | **3.9±0.5** |
| | VIME | 67.0 | 30.2±0.9 | 24.8±0.8 | 15.1±0.6 | 4.7±0.3 | 53.1±0.5 | 34.7±1.4 | **2.9±0.4** | 32.4±1.0 | 28.5±0.9 | 29.7±0.6 | 24.1±0.5 | 51.8±0.6 | 37.9±1.5 | **15.7±0.8** |
| | STG | 66.4 | 63.1±0.6 | 59.5±0.5 | 57.8±0.4 | 56.4±0.3 | 53.9±0.2 | 63.8±0.7 | **46.3±0.6** | 64.5±0.7 | 61.2±0.6 | 60.5±0.4 | 58.7±0.4 | 56.1±0.3 | 64.7±0.9 | **54.4±0.8** |
| | TabNet | 67.4 | 25.1±1.2 | 19.4±0.8 | 13.8±0.7 | 7.1±0.5 | 49.8±0.6 | 34.6±1.2 | **4.7±0.3** | 56.4±1.3 | 49.8±1.2 | 38.7±0.9 | 28.5±0.7 | 47.5±0.8 | 41.3±1.5 | **13.1±0.9** |
| CTU | TabTr. | 95.3 | 95.3±0.0 | 95.3±0.0 | 95.3±0.0 | 95.3±0.0 | 95.3±0.0 | 95.3±0.0 | 95.3±0.0 | 95.3±0.0 | 95.3±0.0 | 95.3±0.0 | 95.3±0.0 | 95.3±0.0 | 95.3±0.0 | 95.3±0.0 |
| | RLN | 97.8 | 98.1±0.1 | 97.4±0.1 | 98.0±0.1 | 97.8±0.1 | 97.6±0.1 | 96.2±0.4 | **84.9±0.6** | 95.5±0.3 | 94.8±0.6 | 95.1±0.2 | 93.0±0.3 | 96.4±0.1 | 94.8±0.5 | **88.7±0.7** |
| | VIME | 95.1 | 95.1±0.1 | 94.2±0.1 | 95.0±0.1 | 94.9±0.1 | 94.8±0.1 | 87.6±2.8 | **14.5±1.2** | 71.4±0.8 | 65.2±0.9 | 50.1±1.0 | 38.5±0.7 | 92.1±0.3 | 71.8±2.1 | **22.4±1.0** |
| | STG | 95.3 | 95.3±0.0 | 95.3±0.0 | 95.3±0.0 | 95.3±0.0 | 95.3±0.0 | 95.3±0.0 | 95.3±0.0 | 95.3±0.0 | 95.3±0.0 | 95.3±0.0 | 95.3±0.0 | 95.3±0.0 | 95.3±0.0 | 95.3±0.0 |
| | TabNet | 96.1 | 95.9±0.2 | 95.7±0.1 | 95.8±0.1 | 96.0±0.1 | 84.7±1.8 | 26.1±2.4 | **10.6±0.3** | 14.2±0.7 | 11.5±0.6 | 8.9±0.4 | 6.5±0.3 | 29.1±1.4 | 11.2±0.9 | **2.8±0.2** |
| WIDS | TabTr. | 75.5 | 67.2±0.6 | 65.8±0.3 | 56.5±0.4 | 48.9±0.5 | 68.8±0.3 | 64.7±0.9 | **41.9±0.7** | 65.1±0.8 | 63.5±0.7 | 61.7±0.5 | 55.4±0.4 | 69.4±0.3 | 67.5±1.1 | **48.3±0.8** |
| | RLN | 77.5 | 74.1±0.5 | 73.0±0.6 | 68.9±0.2 | 62.7±0.2 | 76.9±0.2 | 71.8±0.7 | **49.5±0.5** | 77.4±0.5 | 76.5±0.4 | 66.2±0.7 | **55.2±0.4** | 78.1±0.2 | 73.5±0.8 | 56.8±0.6 |
| | VIME | 72.3 | 68.9±0.7 | 66.5±0.6 | 60.8±0.5 | 53.6±0.1 | 71.5±0.3 | 64.1±1.0 | **40.6±0.8** | 68.2±0.7 | 67.1±0.8 | 62.9±0.5 | 59.1±0.5 | 70.8±0.3 | 67.8±1.1 | **50.3±0.7** |
| | STG | 77.6 | 76.8±0.4 | 75.5±0.8 | 72.1±0.3 | **61.9±0.6** | 77.2±0.2 | 73.5±0.6 | 67.6±0.4 | 71.9±0.5 | 70.8±1.0 | **63.3±0.4** | 65.1±0.4 | 72.1±0.2 | 69.5±0.8 | 64.1±1.2 |
| | TabNet | 79.7 | 27.5±0.8 | 24.8±0.8 | 15.4±0.5 | 10.9±0.6 | 74.8±0.5 | 36.1±1.5 | **1.2±0.2** | 23.4±0.9 | 21.5±1.2 | 12.5±0.6 | 7.8±0.5 | 48.9±1.0 | 16.2±1.2 | **4.1±0.5** |
| MALWARE | TabTr. | 93.6 | 93.1±0.6 | **82.7±0.8** | 90.4±0.5 | 89.4±0.3 | 93.3±0.2 | 91.8±0.7 | 86.3±0.5 | 92.4±0.7 | 91.8±0.6 | 90.2±0.5 | 89.7±0.4 | 93.4±0.3 | 92.5±0.8 | **87.9±0.6** |
| | RLN | 95.0 | 94.7±0.5 | 94.1±0.4 | 92.9±0.4 | 91.8±0.3 | 94.6±0.2 | 93.5±0.6 | **88.9±0.4** | 93.8±0.6 | 93.5±0.5 | **89.3±0.3** | 92.1±0.4 | 94.7±0.2 | 94.2±0.7 | 92.4±0.5 |
| | VIME | 95.0 | 94.6±0.4 | 94.3±0.4 | 93.1±0.3 | 92.5±0.2 | 94.9±0.2 | 94.0±0.5 | **89.8±0.6** | 94.1±0.5 | 93.8±0.6 | 92.5±0.4 | 92.0±0.4 | 94.4±0.2 | 93.9±0.6 | **90.7±0.7** |
| | STG | 93.0 | 91.8±0.6 | 90.8±0.5 | 88.7±0.5 | 89.5±0.4 | 92.5±0.3 | 90.1±0.7 | **85.4±0.6** | 90.9±0.6 | 90.2±0.7 | 89.8±0.4 | 89.3±0.4 | 92.3±0.3 | 91.2±0.8 | **87.5±0.8** |
| | TabNet | 99.0 | 98.7±0.3 | 98.4±0.2 | 97.2±0.2 | 96.5±0.1 | 98.8±0.1 | 97.9±0.4 | **94.1±0.3** | 98.4±0.3 | 98.1±0.2 | 97.3±0.2 | 96.9±0.2 | 98.5±0.1 | 98.2±0.5 | **95.4±0.5** |

and Cyclic. As detailed in Table 1, each dataset comprises four continuous features generated via distinct structural equations, incorporating both linear and non-linear mechanisms and a binary target label, with intrinsic Gaussian noise injected to simulate realistic stochastic dependencies. For each topology, we generate a total of 10,000 samples, strictly split into 8,000 for training and 2,000 for testing.

**Architectures.** Following Simonetto et al. (2024a), we test five widely used architectures: TabTransformer (TabTr.) (Huang et al., 2020), TabNet (Arik & Pfister, 2021), RLN (Shavitt & Segal, 2018), STG (Yamada et al., 2020) and VIME (Yoon et al., 2020).

**Baselines.** We benchmark LCSA against widely adopted adversarial attacks in the tabular domain. The comparison includes **gradient-based baselines** such as PGD (Madry et al., 2017), CaFA (Ben-Tov et al., 2024), and CAPGD (Simonetto et al., 2024c), as well as **generative manifold attacks** such as Adv-VAE (He et al., 2025c). Additionally, we include **heuristic search-based baselines** like BF* (Kulynych et al., 2018) and MOEVA (Simonetto et al., 2021). Note that we exclude ensemble frameworks (e.g., CAA (Simonetto et al., 2024a)), as their underlying constituent attacks are already covered individually in our comparison.

**Evaluation metrics.** Effectiveness is measured by robust accuracy, i.e., classification accuracy on valid adversarial samples. Unchanged misclassified samples remain unperturbed and invalid adversarial examples are counted as correct.

**Perturbation Budgets.** We enforce two constraints to ensure validity. (1) **Geometric Budget ($\epsilon$):** We normalize features to $[0,1]$ and set the $\ell_\infty$ or $\ell_2$ budget $\epsilon \in [0,1]$. (2) **Structural Budget ($\gamma$):** To adapt to varying likelihood scales across datasets, we employ a **data-driven quantile strategy**. We set $\gamma$ to the $q$-th percentile (e.g., $q = 0.95$) of the training set's structural violation scores.

**Supplement.** All datasets, architectures, baselines and implementation details are in **Appendix E**

## 4.2. Overall Performance (*RQ1*)

We first evaluate the robust accuracy of LCSA in comparison with other methods. To ensure a rigorous comparison, we standardize the geometric constraints with a fixed budget of $\epsilon = 0.5$ for both $L_2$ and $L_\infty$ norms, alongside a structural budget $\gamma = 0.5$ corresponding to the 0.5-quantile (50th percentile). As shown in Table 2, LCSA achieves the lowest robust accuracy in **23 out of 25** cases under the $L_2$ constraint and **22 out of 25** under the $L_\infty$ constraint. On the URL dataset, LCSA obliterates model performance, reducing the robust accuracy of TabTransformer from 93.6% (clean) to just 2.1% ($L_2$) and 3.0% ($L_\infty$), significantly widening the performance gap compared to CAPGD. Furthermore, LCSA demonstrates superior stability, maintaining consistently low standard deviations across diverse settings. These results verify the effectiveness and reliability of LCSA as an adversarial attack tool, capable of identifying deeper vulnerabilities even under stringent structural constraints.

## 4.3. Structural Consistency Study (*RQ1*)

We evaluate the mechanistic validity of LCSA using the synthetic benchmarks defined in Table 1. To rigorously quantify whether adversarial perturbations respect the predefined data-generating relationships, we introduce the Structural Consistency Error (SCE). For an adversarial example $\tilde{x} \in \mathbb{R}^d$, SCE measures the mean deviation from the ground-truth structural equations:

$$\text{SCE}(\tilde{x}) = \frac{1}{d} \sum_{j=1}^{d} \| \tilde{x}_j - f_j^*(\tilde{x}_{\text{PA}_j}) \|_2, \quad (20)$$

*Table 3.* Structural Consistency Analysis on Synthetic Structural Benchmarks. We evaluate attacks under geometric constraints ($L_2 = 0.5$, $L_\infty = 0.5$). Results are reported as **"Robust Acc. / SCE"**, where **SCE (Structural Consistency Error)** measures the mean deviation from the predefined data-generating functions. Lower SCE indicates better structural validity

| Structural Topology | $L_2 = 0.5$ | | | | | | | $L_\infty = 0.5$ | | | | | |
| --- | --- | --- | --- | --- | --- | --- | --- | --- | --- | --- | --- | --- | --- |
| | PGD | Adv-VAE | CaFA | CAPGD | BF* | MOEVA | LCSA | PGD | Adv-VAE | CaFA | CAPGD | BF* | MOEVA | LCSA |
| **Chain** | 12.5 / 1.42 | 14.9 / 1.28 | 18.7 / 0.80 | **8.3** / 1.59 | 88.4 / 0.92 | 41.6 / 1.15 | 13.9 / **0.14** | 7.2 / 1.68 | 9.5 / 1.43 | 14.2 / 0.98 | **4.8** / 1.72 | 82.1 / 1.12 | 35.4 / 1.36 | 9.5 / **0.05** |
| **Collider** | 25.4 / 2.15 | 27.1 / 1.92 | 31.8 / 1.05 | 19.5 / 2.28 | 90.9 / 1.58 | 54.7 / 1.82 | **16.2** / **0.11** | 18.5 / 2.42 | 21.4 / 2.18 | 26.9 / 1.24 | 14.2 / 2.55 | 85.7 / 1.76 | 48.2 / 2.05 | 10.8 / **0.09** |
| **Confounder** | 18.6 / 1.92 | 20.9 / 1.78 | 25.3 / 0.95 | **14.8** / 2.01 | 88.2 / 1.37 | 49.4 / 1.68 | 16.7 / **0.08** | 12.4 / 2.15 | 15.8 / 1.94 | 16.1 / 1.15 | **9.1** / 2.27 | 84.2 / 1.46 | 42.6 / 1.85 | 12.7 / **0.07** |
| **Cyclic** | 22.4 / 2.06 | 25.1 / 1.92 | 29.8 / 1.47 | 18.7 / 2.15 | 88.5 / 1.60 | 54.2 / 1.83 | **15.3** / **0.64** | 15.6 / 2.31 | 17.9 / 2.15 | 20.8 / 1.68 | 12.1 / 2.42 | 83.2 / 1.77 | 47.6 / 2.02 | 10.3 / **0.31** |

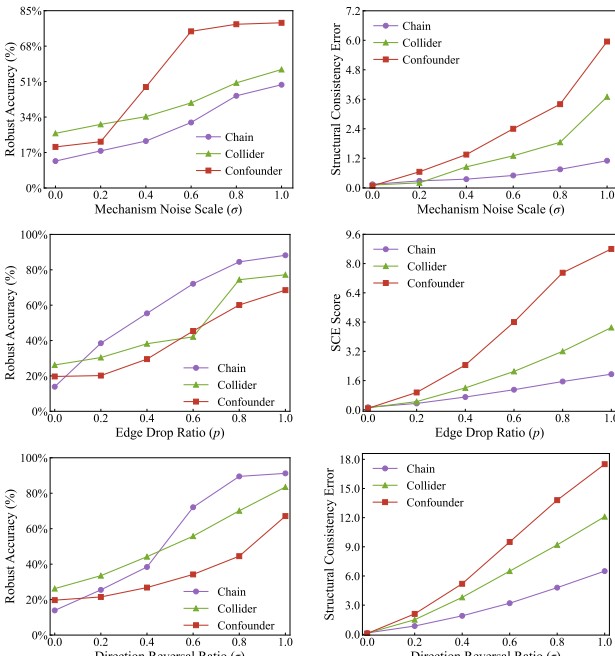

*Figure 2.* Evaluation of robust accuracy under varying $L_2$ and $\gamma$ perturbation budgets on LCSA attack

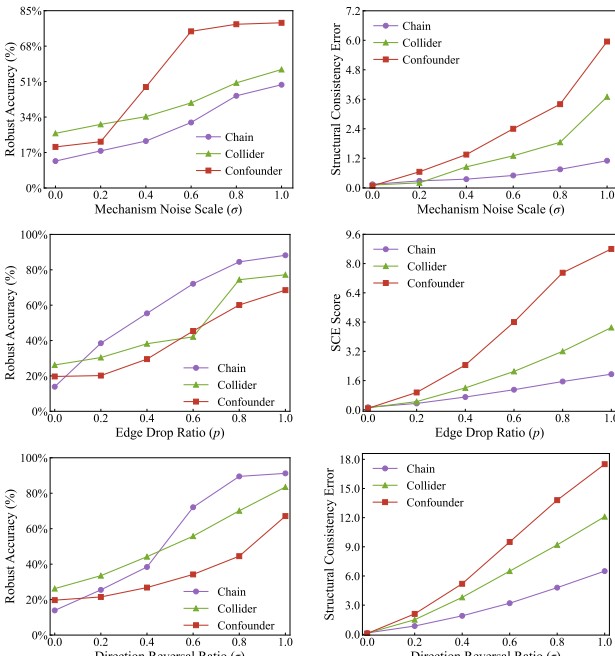

*Figure 3.* Evaluation of robust accuracy and SCE under varying **mechanism noise scale $\sigma$, edge drop ratio $p$ and direction reversal ratio $\tau$** on LCSA attack

where $d$ is the feature dimensionality, $f_j^*$ denotes the predefined data-generating function for feature $j$ (e.g., $x_2 = 2x_1$), and $\tilde{x}_{\text{PA}_j}$ represents the values of the parent variables of $j$. A lower SCE indicates stricter adherence to the data-generating process. As detailed in Table 3, symmetric correlation-based baselines like CAPGD achieve the lowest robust accuracy (e.g., **8.3%** on Chain) but incur massive SCE ($> 1.5$), indicating that they minimize loss by violating the structural relationships of the data. In contrast, **LCSA enforces an additional structural budget** $\gamma = 0.5$ during optimization, which restricts the search space to the admissible region defined by the learned structural model. While this strict constraint leads to a marginal increase in

robust accuracy (**13.9%**), it ensures mechanistic plausibility by reducing SCE by an order of magnitude (e.g., **0.14** on Chain). Remarkably, in the Cyclic topology featuring **feedback loops and non-additive noise**, LCSA employs a forced DAG as a computational scaffold for stable topological ordering, enabling the Ripple mechanism to precondition gradient flows without infinite recursion. As a result, LCSA dominates by achieving both the lowest robust accuracy (**15.3%**) and a severely restricted SCE (**0.64** vs. 2.15 for CAPGD). This validates that ripple mechanism facilitates the optimizer in escaping local optima common in heterogeneous spaces. Furthermore, we extend the structural consistency analysis to real-world datasets in **Appendix I**.

### 4.4. Sensitivity Study of Perturbation Budget (*RQ2*)

We systematically evaluate the sensitivity of LCSA to varying constraints across five datasets. For completeness, the sensitivity analysis for other methods is detailed in **Appendix F**. As illustrated in Fig. 2, we visualize the robust accuracy under a spectrum of structural budgets ($\gamma$, x-axis) and geometric budgets ($L_2$, y-axis). LCSA maintains low robust accuracy even under stringent constraints where perturbation space is severely limited. For instance, on URL dataset, with the strictest budget of $L_2 = 0.1$ and $\gamma = 0.1$, LCSA suppresses robust accuracy to 25.4%. As the budget relaxes to a moderate setting of $L_2 = 0.3$ and $\gamma = 0.3$, the robust accuracy drops to 4.5%, demonstrating LCSA's capability to identify vulnerable structural subspaces.

### 4.5. Robustness Study of SCM Quality (*RQ3*)

To demonstrate the practical reliability of LCSA, we conduct a rigorous stress test against imperfections in the learned SCM under the joint constraint $L_2 = 0.5\,\&\,\gamma = 0.5$. We analyze three error types: (1) **Mechanism Noise ($\sigma$)**, injecting Gaussian perturbations into structural functions; (2) **Structural Sparsity ($p$)**, randomly masking edges in the adjacency matrix; and (3) **Direction Reversal ($\tau$)**, randomly reversing edges' direction in the adjacency matrix.

*Table 4.* Evaluation of robust accuracy in transfer attack between STG, TabNet and TabTr.

| Method | Dataset | TabTr. → STG | TabTr. → TabNet | TabNet → STG | TabNet → TabTr. | STG → TabTr. | STG → TabNet |
|---|---|---|---|---|---|---|---|
| CaFA | URL | 84.5 | 58.2 | 86.1 | 45.3 | 48.2 | 55.6 |
| | LCLD | 63.1 | 38.5 | 64.1 | 55.4 | 58.7 | 35.2 |
| | CTU | 95.3 | 96.0 | 95.3 | 95.3 | 95.3 | 95.9 |
| | WIDS | 74.2 | 35.6 | 75.1 | 71.5 | 70.8 | 32.4 |
| | MALWARE | 91.0 | 98.5 | 91.2 | 92.8 | 93.1 | 98.6 |
| CAPGD | URL | 81.7 | 52.4 | 82.5 | 38.6 | 42.5 | 48.9 |
| | LCLD | 61.5 | 32.1 | 62.8 | 51.9 | 54.1 | 29.3 |
| | CTU | 95.3 | 95.9 | 95.3 | 95.3 | 95.3 | 95.8 |
| | WIDS | 71.8 | 30.5 | 73.4 | 68.7 | 66.5 | 28.1 |
| | MALWARE | 90.2 | 98.2 | 89.8 | 92.1 | 92.5 | 98.3 |
| MOEVA | URL | 89.4 | 76.5 | 88.9 | 65.2 | 68.4 | 79.1 |
| | LCLD | 66.2 | 48.5 | 67.1 | 63.8 | 62.5 | 45.2 |
| | CTU | 95.3 | 95.9 | 95.3 | 95.3 | 95.3 | 95.9 |
| | WIDS | 76.5 | 62.4 | 75.8 | 72.4 | 71.5 | 58.9 |
| | MALWARE | 92.5 | 98.6 | 91.8 | 93.4 | 93.8 | 98.5 |
| LCSA | URL | 75.6 | 45.8 | 78.2 | 25.4 | 35.6 | 41.2 |
| | LCLD | 55.4 | 22.8 | 58.6 | 42.4 | 46.8 | 18.5 |
| | CTU | 95.3 | 95.8 | 95.3 | 95.3 | 95.3 | 95.7 |
| | WIDS | 65.7 | 22.4 | 68.5 | 59.8 | 58.2 | 18.6 |
| | MALWARE | 88.5 | 97.5 | 88.1 | 90.5 | 91.2 | 97.8 |

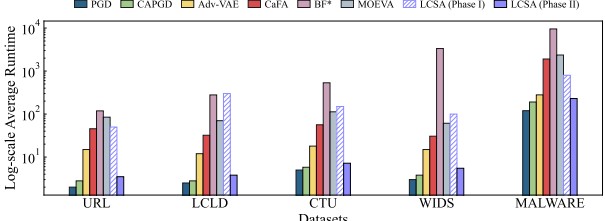

PGD · CAPGD · Adv-VAE · CaFA · BF* · MOEVA · LCSA (Phase I) · LCSA (Phase II)

*Figure 4.* Runtime comparison (log scale) across five tabular datasets. We decouple LCSA into Phase I and Phase II

**Operational Stability under Mechanism Noise.** We verify whether LCSA remains effective under realistic estimation errors. We first quantified the intrinsic epistemic uncertainty of our SCMs, finding the average empirical residual noise to be approximately $\hat{\sigma}_{\text{real}} \approx 0.12$. As shown in Fig. 3, the **Confounder** topology maintains a low Robust Accuracy of 22.2% at $\sigma = 0.1$, and degrades gracefully only after $\sigma > 0.4$. This implies a **safety margin of nearly** $4\times$ between real-world noise levels and the breakdown point.

**Tolerance to Structural Imperfections.** We further evaluate LCSA's resilience to missing structural links (Fig. 3). While severe edge removal eventually hinders gradient flow, LCSA demonstrates considerable tolerance. For the **Confounder** topology, even with 40% of edges removed ($p = 0.4$), the Robust Accuracy remains low at 29.5%. Although the **Chain** topology is more sensitive to connectivity due to its sequential gradient path, it retains strong attack capabilities at moderate sparsity levels ($p \leq 0.2$).

**Robustness to Directional Misalignment.** Finally, we evaluate LCSA under edge reversal. For the **Confounder** topology, the attack remains consistently effective, with Robust Accuracy staying below 35% even as the SCE rises to 9.2. Similarly, the **Chain** topology maintains high attack success ratio under moderate perturbations ($\tau < 0.4$) before degrading. These results confirm that LCSA possesses a significant safety margin: it does not demand a flawless SCM and can reliably effectively exploit approximate causal structures to guide attacks under realistic noise levels.

### 4.6. Time Efficiency Study (*RQ4*)

We evaluate the runtime efficiency of LCSA against six baseline attacks across five datasets. As illustrated in Fig. 4,

*Table 5.* Evaluation of robust accuracy in adversarial training

| Dataset | Model | Clean | PGD | Adv-VAE | CaFA | CAPGD | BF* | MOEVA | LCSA |
|---|---|---|---|---|---|---|---|---|---|
| URL | $AT_{TabTr.}$ | 91.5 | 89.3 | 88.1 | 85.4 | 86.7 | 90.9 | 84.2 | 88.9 |
| | $AT_{RLN}$ | 92.8 | 91.4 | 90.2 | 86.8 | 88.7 | 92.0 | 85.5 | 90.6 |
| | $AT_{VIME}$ | 90.4 | 88.9 | 87.2 | 82.7 | 85.4 | 89.6 | 81.5 | 88.1 |
| | $AT_{STG}$ | 91.2 | 90.3 | 89.5 | 84.2 | 86.5 | 90.7 | 83.8 | 88.5 |
| | $AT_{TabNet}$ | 91.8 | 90.7 | 89.4 | 85.6 | 87.2 | 91.1 | 84.5 | 89.8 |
| LCLD | $AT_{TabTr.}$ | 67.2 | 65.4 | 64.2 | 58.7 | 62.1 | 66.3 | 60.5 | 65.1 |
| | $AT_{RLN}$ | 65.8 | 64.5 | 63.4 | 60.2 | 62.5 | 65.1 | 61.5 | 63.8 |
| | $AT_{VIME}$ | 64.5 | 62.6 | 61.2 | 55.8 | 60.1 | 63.5 | 58.2 | 61.5 |
| | $AT_{STG}$ | 64.2 | 62.8 | 61.5 | 59.5 | 60.8 | 63.4 | 59.1 | 62.1 |
| | $AT_{TabNet}$ | 65.1 | 63.5 | 62.4 | 56.5 | 60.5 | 64.2 | 60.4 | 62.9 |
| CTU | $AT_{TabTr.}$ | 95.3 | 95.3 | 95.3 | 95.3 | 95.3 | 95.3 | 95.3 | 95.3 |
| | $AT_{RLN}$ | 97.2 | 96.6 | 96.3 | 95.6 | 96.1 | 96.9 | 95.6 | 97.1 |
| | $AT_{VIME}$ | 93.8 | 92.7 | 91.5 | 88.9 | 90.4 | 93.0 | 89.5 | 92.4 |
| | $AT_{STG}$ | 95.3 | 95.3 | 95.3 | 95.3 | 95.3 | 95.3 | 95.3 | 95.3 |
| | $AT_{TabNet}$ | 95.6 | 94.5 | 93.2 | 85.6 | 91.8 | 95.1 | 88.7 | 94.7 |
| WIDS | $AT_{TabTr.}$ | 73.8 | 72.8 | 71.5 | 68.2 | 70.4 | 73.1 | 69.4 | 71.8 |
| | $AT_{RLN}$ | 75.1 | 74.3 | 73.4 | 70.4 | 72.6 | 74.6 | 71.8 | 73.9 |
| | $AT_{VIME}$ | 69.2 | 67.9 | 67.1 | 64.3 | 66.5 | 68.6 | 65.5 | 68.4 |
| | $AT_{STG}$ | 75.4 | 74.2 | 73.1 | 70.5 | 72.8 | 75.0 | 71.5 | 74.1 |
| | $AT_{TabNet}$ | 77.2 | 76.1 | 74.7 | 65.8 | 71.5 | 76.4 | 68.2 | 75.5 |
| MALWARE | $AT_{TabTr.}$ | 92.1 | 91.5 | 91.1 | 90.8 | 90.4 | 91.7 | 90.1 | 91.6 |
| | $AT_{RLN}$ | 94.2 | 93.7 | 93.4 | 93.1 | 93.5 | 93.9 | 92.8 | 93.9 |
| | $AT_{VIME}$ | 93.5 | 93.8 | 92.2 | 92.3 | 92.6 | 93.2 | 92.4 | 93.3 |
| | $AT_{STG}$ | 91.5 | 91.0 | 90.7 | 90.2 | 90.4 | 91.3 | 89.6 | 91.2 |
| | $AT_{TabNet}$ | 97.5 | 97.2 | 96.9 | 96.4 | 96.8 | 97.3 | 96.5 | 97.4 |

*Table 6.* Transferability from TabNet to non-differentiable GBDTs on the URL dataset ($L_2 = 0.5\,\&\,\gamma = 0.5$). Clean accuracies for XGBoost and LightGBM are 97.4% and 98.2%, respectively

| Method | TabNet → XGBoost | TabNet → LightGBM |
|---|---|---|
| PGD | 78.5 | 81.2 |
| Adv-VAE | 49.4 | 52.8 |
| CaFA | 58.7 | 61.9 |
| CAPGD | 51.2 | 64.5 |
| BF* | 85.3 | 83.7 |
| MOEVA | 37.1 | 49.6 |
| **LCSA** | **32.4** | **35.1** |

the results are presented on a logarithmic scale to accommodate the significant disparity between gradient-based and heuristic methods. LCSA strategically decouples structural discovery from adversarial optimization. While Phase I incurs a one-time computational cost for structural learning, this overhead is amortized during deployment. In the critical Phase II, LCSA demonstrates exceptional efficiency. Despite incorporating structural constraints (via forward SCM passes and soft-embedding), LCSA achieves inference speeds comparable to state-of-the-art baselines. This confirms that LCSA successfully enforces semantic realism without compromising real-time attack feasibility.

### 4.7. Scalability Study (*RQ5*)

We analyze the performance of LCSA in two critical scenarios: defense through adversarial training (AT) and offense via cross-model transferability, under the joint constraints of perturbation bound $L_2 = 0.5$ and parameter $\gamma = 0.5$.

**(1) Robustness against AT.** We adopt the framework of Madry et al. (2017) to evaluate robustness. As shown in Table 5, models trained with LCSA demonstrate remarkable resilience. On the URL dataset, the robust accuracy against LCSA recovers to 88.9% for TabTr. and 90.6% for RLN, approaching clean levels. This defense generalizes to unseen attacks including CaFA and CAPGD (yielding 85.4% and 86.7% respectively for the AT-enhanced TabTr.), confirming the effective hardening of decision boundaries.

*Table 7.* Anomaly Detection Ratio (%) of successful adversarial examples ($L_2 = 0.5$ & $\gamma = 0.5$). Lower values indicate higher stealthiness and better evasion of the anomaly detector

| Method | URL | LCLD | CTU | WIDS | MALWARE |
|---|---|---|---|---|---|
| PGD | 70.2 | 58.1 | 98.3 | 90.4 | 75.4 |
| Adv-VAE | 48.1 | 59.4 | 79.8 | 92.7 | 55.3 |
| BF* | 85.6 | 77.8 | 98.7 | 84.8 | 89.1 |
| MOEVA | 38.3 | 26.5 | 70.2 | 60.6 | 48.3 |
| CaFA | 32.7 | 21.3 | 73.7 | 63.2 | 39.7 |
| CAPGD | 61.4 | 49.6 | 85.6 | 78.5 | 62.4 |
| LCSA | **19.4** | **10.3** | **52.9** | **33.7** | **23.2** |

**(2) Cross-Model Transferability.** Table 4 benchmarks black-box transferability against competitive baselines, where LCSA consistently achieves the lowest target robust accuracy. When transferring from TabTr. to TabNet on the URL dataset, LCSA reduces accuracy to 45.8%, outperforming CaFA (58.2%) and MOEVA (76.5%). A similar reduction to 18.5% occurs for the STG to TabNet transfer on the LCLD dataset, confirming the ability of LCSA to capture generalized, architecture-agnostic structural vulnerabilities. Although white-box generation requires differentiable models (e.g., TabNet), LCSA produces structurally plausible perturbations that transfer effectively to **non-differentiable tree ensembles.** Table 6 shows that transferring adversarial examples generated on TabNet using the URL dataset to XGBoost and LightGBM allows LCSA to significantly outperform all baselines, reducing robust accuracy to **32.4%** and **35.1%** respectively. Unlike the competitive baseline MOEVA, LCSA avoids prohibitive black-box evolutionary searches (requiring approximately $10^3$ seconds), thereby reducing generation time by orders of magnitude. GBDTs learn joint decision rules based on feature co-occurrences. Standard smooth perturbations shift features in isolation and often fail to cross orthogonal, axis-aligned hard splits. In contrast, the proposed ripple mechanism coherently shifts features along correlated structural paths. Upon discretization, this coordinated movement systematically bypasses joint decision boundaries. Rather than falling into sparse, isolated leaves that trees natively reject, the structurally plausible ripple directs samples into valid, dense target regions.

### 4.8. Stealthiness against Anomaly Detectors (*RQ6*)

To evaluate the practical threat of LCSA in real-world deployment scenarios, we measure the stealthiness of the method against standard anomaly detection pipelines. We simulate a standard defense pipeline by training an Isolation Forest (Liu et al., 2008) exclusively on benign training data for each dataset. Subsequently, we aggregate the successfully generated adversarial examples across all five target architectures. We report the *Anomaly Detection Ratio*, defined as the percentage of successful adversarial examples flagged as anomalies by the detector. Table 7 shows that although standard gradient-based methods such as CAPGD and CaFA achieve competitive attack success rates, the structure-blind perturbations of these methods generate out-of-distribution

samples that real-world detectors easily intercept. For instance, on the URL dataset, CAPGD and CaFA trigger the detector at rates of 61.4% and 32.7%, respectively. Similarly, heuristic searches such as BF* exhibit severe detection rates (e.g., 98.7% on the CTU dataset), rendering the methods impractical in guarded systems. LCSA consistently exhibits exceptional evasion capabilities across all benchmarks, yielding the lowest anomaly detection ratios (e.g., **19.4%** on the URL dataset and **10.3%** on the LCLD dataset). This empirical evidence corroborates the core hypothesis: by optimizing within the structural constraints delineated by the learned SCM, the ripple mechanism ensures that the perturbations of LCSA are both statistically likely and semantically consistent with the underlying data manifold.

### 4.9. *Other Analyses & Related Work*

To address RQ7, we present detailed experimental results in **Appendix G** and **H**. Additionally, a comprehensive review of related work is provided in **Appendix A**.

## 5. Limitations

Although LCSA advances robustness evaluation, it introduces dual-use risks in high-stakes tabular domains (e.g., finance, healthcare, and fraud detection), where adversaries could weaponize the resulting highly transferable and structurally consistent perturbations. Our primary objective is to expose these vulnerabilities to catalyze resilient defenses. As the adversarial training results demonstrate, training against LCSA effectively restores the robustness of decision boundaries, enabling robust accuracy to approach clean performance. This defense also generalizes to unseen attacks such as CaFA and CAPGD. These findings highlight the inadequacy of standard geometric defenses for heterogeneous data. Future work will focus on designing structure-aware validation pipelines that integrate causal dependency tracking directly into model deployment, enabling systems to detect and reject structurally plausible manipulations.

## 6. Conclusion

In this work, we addressed the critical blind spot of tabular adversarial attacks: their tendency to generate perturbations that are numerically valid but semantically inconsistent. We proposed LCSA, a framework that shifts the paradigm from geometric proximity to structural plausibility. By integrating heterogeneous SCMs with a novel structure-aware ripple mechanism, LCSA ensures that adversarial interventions respect the logical dependencies of the tabular data. Empirical results on five real-world and three synthetic datasets demonstrate that LCSA consistently outperforms baselines in 45 out of 50 settings. These findings highlight the necessity of structural priors into adversarial modeling, paving the way for more rigorous evaluations of tabular robustness.

## Acknowledgements

This work was supported by Ant Group Research Fund.

## Impact Statement

This paper presents work whose goal is to advance the field of Machine Learning. There are many potential societal consequences of our work, none which we feel must be specifically highlighted here.

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

# A. Related Work

**Adversarial Attacks on Tabular Data.** Generating adversarial examples for tabular data presents unique challenges due to feature heterogeneity and discrete constraints. Existing white-box methods typically enforce validity via penalty functions (Ballet et al., 2019; Nobi & Krishnan, 2022; Simonetto et al., 2021; 2024a) or adaptive steps (Ben-Tov et al., 2024; Bostani et al., 2022; Kulynych et al., 2018; Sheatsley et al., 2020; 2021; Tian et al., 2020), while black-box approaches rely on heuristics (Abusnaina et al., 2019; Cartella et al., 2021; Dong et al., 2020; Kireev et al., 2022; Mathov et al., 2022) or multi-objective evolutionary algorithms (Ghamizi et al., 2020; Simonetto et al., 2021; Wang et al., 2023). Recently, efficient decision-based attacks have also been proposed to navigate these structured boundaries in black-box settings (Kazoom et al., 2025). To ensure domain consistency, subsequent works introduced explicit constraints, relying on handcrafted logical rules or immutability specifications (He et al., 2025b; Simonetto et al., 2021; 2024a;b). These methods often employ direct projection operators to strictly respect feature bounds and types. However, such manual specifications are inherently labor-intensive. Consequently, recent advancements have shifted towards automated constraint mining (Ben-Tov et al., 2024; Sheatsley et al., 2020; 2021). These approaches attempt to define feasible regions without extensive manual intervention. Despite these improvements, current optimization strategies largely treat features as independent dimensions or rely on symmetric metrics (Dyrmishi et al., 2025; Simonetto et al., 2024b; Wawrowski et al., 2025). This geometric focus creates a critical blind spot: it permits perturbations that maximize classification error but reside in low-density, unrealistic regions of the data manifold. In contrast, LCSA overcomes this limitation by integrating a structural likelihood constraint, forcing perturbations to respect asymmetric dependencies and ensuring statistical coherence in high-stakes domains.

**Semantic Consistency and Causal Structure.** Preserving semantic consistency in tabular adversarial attacks remains an open challenge due to the lack of explicit spatial structure. Prior attempts have largely relied on heuristics, such as restricting perturbations on class-critical features (Chen et al., 2020; Sun et al., 2023) or imposing statistical bounds to maintain distributional similarity (Mathov et al., 2022). Notably, recent work by He et al. (2025a;c) explicitly addresses imperceptibility by crafting on-manifold perturbations via Variational Autoencoders. Other approaches leverage interpretability tools to guide realistic modifications (Author, 2025; Cartella et al., 2021; Shirazi et al., 2021). However, these methods often depend on implicit definitions of plausibility or require extensive domain knowledge (Kireev et al., 2022). Crucially, while generative approaches like (He et al., 2025c) capture associative distributions, they may overlook the directed nature of feature interactions. To provide a more rigorous grounding for data semantics, Causal Structure Learning offers a principled alternative by modeling the generative mechanisms via Directed Acyclic Graphs (Bello et al., 2022; Johansson, 2024; Zheng et al., 2018). While structural priors have been successfully applied to enhance tabular model generalization (Hollmann et al., 2022; 2025; Robertson et al., 2025a) or generate valid counterfactuals (Robertson et al., 2025b), their integration into adversarial robustness remains underexplored. LCSA addresses this gap by connecting semantic consistency with learned structural inference. Unlike previous methods that treat semantics as static rules **or undirected manifolds**, we define valid perturbations as those adhering to the asymmetric dependencies of a learned Causal Structural Model, thereby enforcing structurally consistent downstream effects without manual intervention.

# B. Algorithm

Algorithm 1 summarizes the complete workflow of LCSA. Phase I (Lines 1-5) establishes the structural constraints by learning an ensemble of heterogeneous SCMs. Phase II (Lines 6-24) executes the adversarial optimization over the intervention variable $\nu$. Crucially, in each iteration, we structurally propagate the intervention to obtain the raw perturbation (Eq. 14) and then project it onto the feasible set $\Omega$ to ensure geometric validity. The optimization employs a two-stage Dual-Ascent strategy. Finally, a post-optimization verification ensures that the discretized output $x_{\text{adv}}$ successfully fools the model while satisfying the structural budget $\gamma$.

# C. Complexity Analysis

We analyze the computational complexity of LCSA in two stages: the offline learning of the Heterogeneous Ensemble SCM and the generation of structural-aware adversarial examples. Let $N$ denote the number of training samples, $d$ the number of features, and $M$ the ensemble size.

**Phase I: Heterogeneous Ensemble SCM Learning.** The training involves optimizing $M$ independent structural models. In each iteration, two primary computations occur: *(1) Mechanism Learning:* We update $d$ MLPs for each of the $M$ models. Let $C_{\text{MLP}}$ denote the average complexity of a forward-backward pass for a single MLP. The cost over the dataset is $\mathcal{O}(M \cdot N \cdot d \cdot C_{\text{MLP}})$. *(2) Structure Learning:* The primary computational bottleneck lies in the acyclicity constraint

---

**Algorithm 1** Likelihood-Constrained Structural Attack (LCSA)

---

**Require:** Target model $h_\theta$, benign sample $(x, y)$, training dataset $\mathcal{D}_{data}$
**Require:** Ensemble size $M$, perturbation budget $\epsilon$ ($L_p$-norm)
**Require:** Structural budget $\gamma$, penalty $\rho$, learning rate $\eta$
**Require:** Iterations $T_{\text{warm}}, T_{\text{total}}$
**Ensure:** Adversarial example $x_{\text{adv}}$

1: **Phase I: Heterogeneous Ensemble SCM Learning**
2: **for** $m = 1$ **to** $M$ **do**
3:      Sample bootstrap dataset $\mathcal{D}^{(m)}$ from $\mathcal{D}_{data}$
4:      Optimize $\Phi^{(m)}, W^{(m)}$ via heterogeneous loss      $\triangleright$ Minimize (Eq. 8)
5:      $\hat{\sigma}^2_{j,m} \leftarrow \text{Var}(x_j - \hat{x}^{(m)}_j)$ for $j \in \mathcal{I}_{\text{cont}}$      $\triangleright$ Est. Noise (Eq. 11)
6: **end for**
7: **Phase II: Structural-Aware Adversarial Optimization**
8: $\nu^{(0)} \leftarrow \mathbf{0}, \lambda^{(0)} \leftarrow 0, x_{\text{adv}} \leftarrow$ None
9: **for** $t = 0$ **to** $T_{\text{total}} - 1$ **do**
10:      $\delta^{(t)} \leftarrow \text{Ripple}(\nu^{(t)})$      $\triangleright$ Recursive propagation (Eq. 14)
11:      $\delta^{(t)} \leftarrow \Pi_\Omega(\delta^{(t)})$      $\triangleright$ Project onto feasible set (Eq. 16)
12:      $\tilde{x}^{(t)} \leftarrow x + \delta^{(t)}$
13:      **if** $t < T_{\text{warm}}$ **then**
14:          $g^{(t)} \leftarrow \nabla_\nu \mathcal{L}_{\text{cls}}(h_\theta(\tilde{x}^{(t)}), y)$
15:          $\nu^{(t+1)} \leftarrow \nu^{(t)} + \eta \cdot g^{(t)}$      $\triangleright$ Warm Start (Eq. 16)
16:      **else**
17:          Compute $\psi^{(t)} \leftarrow \mathcal{L}_{\text{scm}}(\tilde{x}^{(t)}) - \mathcal{L}_{\text{scm}}(x) - \gamma$      $\triangleright$ Constraint violation (Eq. 15)
18:          $\mathcal{L}_{\text{aug}} \leftarrow \mathcal{L}_{\text{cls}} - \lambda^{(t)}\psi^{(t)} - \frac{\rho}{2} \max(0, \psi^{(t)})^2$
19:          $\nu^{(t+1)} \leftarrow \nu^{(t)} + \eta \nabla_\nu \mathcal{L}_{\text{aug}}$      $\triangleright$ Dual-Ascent Update (Eq. 17)
20:          Update $\psi^{(t+1)}$ using $\tilde{x}(\nu^{(t+1)})$
21:          $\lambda^{(t+1)} \leftarrow \max(0, \lambda^{(t)} + \rho \cdot \psi^{(t+1)})$      $\triangleright$ Update Multiplier (Eq. 18)
22:      **end if**
23:      $\bar{x} \leftarrow \text{Discretize}(\tilde{x}^{(t+1)})$      $\triangleright$ Project categorical vars to vertices
24:      **if** $h_\theta(\bar{x}) \neq y$ and $\mathcal{L}_{\text{scm}}(\bar{x}) - \mathcal{L}_{\text{scm}}(x) \leq \gamma$ **then**
25:          $x_{\text{adv}} \leftarrow \bar{x}$
26:      **end if**
27: **end for**
28: **return** $x_{\text{adv}}$

---

gradient, which typically involves a matrix inversion $\nabla_W \text{tr}((I - W \odot W)^{-1})$ scaling at $\mathcal{O}(d^3)$ due to the dense nature of the inverse matrix. To resolve this and substantiate our scalability claim, we leverage the strict sparsity enforced by candidate screening (where each node has at most $k$ parents). Instead of performing exact matrix inversion, we approximate the gradient-vector products using a **Truncated Neumann Series** expansion (Bello et al., 2022; Lorraine et al., 2020). This technique transforms the expensive inversion into a sequence of $L$ sparse matrix-vector products (SpMVP). Since $W$ has at most $k \cdot d$ non-zero entries, each SpMVP costs $\mathcal{O}(k \cdot d)$. Performing this approximation across the diagonal dimensions yields a complexity of $\mathcal{O}(L \cdot k \cdot d^2)$. Since the series converges rapidly for DAGs, $L$ is a small constant, effectively reducing the complexity to $\mathcal{O}(k \cdot d^2)$. Thus, for $T_{\text{train}}$ iterations, the total time complexity is:

$$\mathcal{T}_{\text{SCM}} = \mathcal{O}\left(T_{\text{train}} \cdot M \cdot (N \cdot d \cdot C_{\text{MLP}} + k \cdot d^2)\right). \tag{21}$$

With the sparsity cap $k$ (e.g., $k = 150$ for MALWARE), the cubic scaling is avoided, making the framework computationally tractable even for massive feature spaces.

**Phase II: Structural-Aware Adversarial Optimization.** During the attack, we optimize the intervention variable $\nu$ over $T_{\text{total}}$ iterations. The complexity is dominated by the **Recursive Ripple Propagation** and **Gradient Backpropagation**.

*(1) Recursive Forward Pass & Projection:* Unlike standard attacks that use additive noise, LCSA constructs the effective perturbation $\delta$ via Structural Ripple Perturbation (Eq. 14). This requires sequentially executing the ensemble mechanisms

following the topological order. Since the learned graph is sparse (bounded by in-degree $k$), evaluating $d$ features across $M$ models incurs a cost of $\mathcal{O}(M \cdot d \cdot C_{\text{MLP}})$, where $C_{\text{MLP}}$ now depends on the small input dimension $k$. Subsequently, $\Pi_\Omega$ costs $\mathcal{O}(|\mathcal{I}_{\text{cont}}|) + \sum_{j \in \mathcal{I}_{\text{cat}}} \mathcal{O}(K_j \log K_j)$ due to simplex projection.

*(2) Gradient Computation:* Computing the gradient $\nabla_\nu \mathcal{L}_{\text{aug}}$ requires backpropagating through the target model and the recursive structural structure. Since the computational graph sparsity is enforced by the candidate mask, the backward pass complexity aligns with the forward pass, $\mathcal{O}(M \cdot d \cdot C_{\text{MLP}})$.

*(3) Target Model Inference:* The target model forward and backward passes cost $\mathcal{O}(C_{\text{target}})$.

Combining these factors, the complexity is: $\mathcal{T}_{\text{attack}} = \mathcal{O}\left(T_{\text{total}} \cdot (C_{\text{target}} + M \cdot d \cdot C_{\text{MLP}})\right)$.

**Remarks.** Although the asymptotic complexity class remains linear regarding $M$ and $d$, the efficiency is significantly bolstered by the sparse candidate screening in Phase I, which limits the connectivity of the causal graph. This ensures that LCSA remains computationally efficient for practical deployment settings.

## D. Theoretical Justification

In Section 1 and 3, we claimed that LCSA's superior performance stems from the *Ripple Mechanism* acting as a structural preconditioner. This section provides a formal gradient flow analysis to substantiate this claim.

**Problem Setup.** Let $\mathcal{L}_{cls}(h_\theta(\tilde{x}), y)$ be the classification loss. We contrast the optimization dynamics: **(1) Standard Optimization (e.g., PGD):** The optimizer updates $\delta$ via $g_{\text{std}} = \nabla_\delta \mathcal{L}_{cls} \in \mathbb{R}^d$. **(2) LCSA Optimization:** The optimizer updates $\nu$ via $g_{\text{lcsa}} = \nabla_\nu \mathcal{L}_{cls} \in \mathbb{R}^d$, where $\delta(\nu)$ is the Ripple mechanism defined in Eq. (14).

**Gradient Dynamics Analysis.** Recall the Ripple formulation in Eq. (14). Assuming features $j = 1, \ldots, d$ follow the DAG's topological order, the perturbation is $\delta_j(\nu) = \nu_j + \frac{1}{M} \sum_{m=1}^{M} (\hat{f}_j^{(m)}(\tilde{x}_{\text{PA}_j}^{(m)}) - \hat{f}_j^{(m)}(x_{\text{PA}_j}^{(m)}))$. By the chain rule, the LCSA gradient relates to the standard gradient via the Jacobian $\mathbf{J} = \frac{\partial \delta}{\partial \nu}$:

$$\underbrace{\nabla_\nu \mathcal{L}_{cls}}_{g_{\text{lcsa}}} = \left(\frac{\partial \delta}{\partial \nu}\right)^\top \underbrace{\nabla_\delta \mathcal{L}_{cls}}_{g_{\text{std}}} = \mathbf{J}^\top g_{\text{std}}. \tag{22}$$

Eq. (22) shows that $\mathbf{J}^\top$ acts as a linear *preconditioner* reshaping the optimization landscape.

**Proposition 1.** *Assume features are re-indexed according to the **consensus topological ordering** derived from the aggregate DAG $\bar{\mathcal{G}}$. Under this ordering, the Jacobian matrix $\mathbf{J}$ is a lower-triangular matrix with unit diagonals.*

*Proof.* Let $\text{PA}_j^{(m)} = \{i \mid W_{i,j}^{(m)} \neq 0\}$ denote the parent set of feature $j$ in the $m$-th SCM and the indices $1, \ldots, d$ adhere to the consensus topological ordering determined by the aggregate adjacency matrix $\bar{W} = \frac{1}{M} \sum W^{(m)}$. This ensures that for any valid parent-child pair $(k, j)$ in the consensus view, we have $k < j$. We analyze the Jacobian entries $[\mathbf{J}]_{ji} = \frac{\partial \delta_j}{\partial \nu_i}$ by differentiating the recursive definition:

1. **Case $j = i$ (Diagonal Terms):** By definition, $\delta_i = \nu_i + \mathcal{S}_i$, where $\mathcal{S}_i = \frac{1}{M} \sum_{m=1}^{M} \left(\hat{f}_i^{(m)}(\tilde{x}_{\text{PA}_i}^{(m)}) - \hat{f}_i^{(m)}(x_{\text{PA}_i}^{(m)})\right)$ represents the structural ensemble term. Applying the chain rule w.r.t. $\nu_i$:

$$\frac{\partial \delta_i}{\partial \nu_i} = \frac{\partial \nu_i}{\partial \nu_i} + \sum_{k \in \cup_m \text{PA}_i^{(m)}} \frac{\partial \mathcal{S}_i}{\partial \tilde{x}_k} \frac{\partial \tilde{x}_k}{\partial \nu_i}. \tag{23}$$

   Note that $\tilde{x}_k = x_k + \delta_k$. Since valid parent nodes $k$ precede $i$ in the consensus topological order ($k < i$), their perturbations $\delta_k$ are functionally independent of $\nu_i$ (i.e., $\frac{\partial \delta_k}{\partial \nu_i} = 0$). Consequently, $\frac{\partial \delta_i}{\partial \nu_i} = 1$.

2. **Case $j < i$ (Upper Triangle):** The perturbation $\delta_j$ is constructed recursively following the consensus order. Mathematically, $\delta_j$ is a function solely of the latent variables $\{\nu_1, \ldots, \nu_j\}$ from preceding nodes. Since $i > j$, $\nu_i \notin \{\nu_1, \ldots, \nu_j\}$, implying strict independence. Thus, $\frac{\partial \delta_j}{\partial \nu_i} = 0$.

3. **Case $j > i$ (Lower Triangle - Downstream Propagation):** If $j$ is a downstream node, $\delta_j$ depends on the perturbations

*Table 8.* Dataset Statistics

| Dataset | Domain | Output to flip | Total size | # Features | # Constraints | Balance (%) |
|---------|--------|----------------|-----------:|-----------:|--------------:|-------------|
| CTU | Botnet detection | Malicious connections | 198 128 | 756 | 360 | 99.3/0.7 |
| LCLD | Credit scoring | Reject loan request | 1 220 092 | 28 | 9 | 80/20 |
| URL | Phishing | Malicious URL | 11 430 | 63 | 14 | 50/50 |
| MALWARE | Malware detection | Malicious software | 17 584 | 24 222 | 7 | 45.5/54.5 |
| WIDS | ICU survival | Expected survival | 91 713 | 186 | 31 | 91.4/8.6 |

of its parents. Applying the chain rule to the ensemble expectation:

$$\frac{\partial \delta_j}{\partial \nu_i} = \frac{1}{M} \sum_{m=1}^{M} \sum_{k \in \mathrm{PA}_j^{(m)}} \underbrace{\left( \frac{\partial \hat{f}_j^{(m)}}{\partial \tilde{x}_k} \cdot \frac{\partial \delta_k}{\partial \nu_i} \right)}_{\text{Model } m \text{ Contribution}}. \tag{24}$$

**Handling Structural Conflicts:** Since the SCMs are heterogeneous, a specific model $m$ might contain a noisy edge $(k, j)$ where $k > j$ (violating the consensus order). However, the forward propagation of the Ripple Mechanism is strictly executed according to the consensus order. Mathematically, this enforces that for any $k > j$, the dependency is effectively severed in the computation graph of the current step (i.e., $\frac{\partial \delta_k}{\partial \nu_i}$ is treated as 0 for this path). Consequently, the summation only accumulates gradients along structurally valid paths consistent with the consensus DAG. The averaging operator $\frac{1}{M} \sum$ thus acts as a structural denoising filter: consistent structural dependencies are reinforced, while conflicting or spurious edges are suppressed. The Jacobian entry is non-zero if and only if there is a valid path from $i$ to $j$ in the consensus view.

Since $[\mathbf{J}]_{ji} = 0$ for all $j < i$, $\mathbf{J}$ is lower-triangular. The element-wise update rule becomes:

$$[\nabla_\nu \mathcal{L}_{cls}]_i = [\nabla_\delta \mathcal{L}_{cls}]_i + \sum_{j \in \mathrm{Descendants}(i)} \frac{\partial \delta_j}{\partial \nu_i} \cdot [\nabla_\delta \mathcal{L}_{cls}]_j. \tag{25}$$

**Implication:** Eq. (25) justifies the resolution of the Optimization Paradox. In **Standard PGD**, optimization often stagnates due to *Gradient Masking*, where the direct gradient is negligible ($[\nabla_\delta \mathcal{L}_{cls}]_i \approx 0$) in heterogeneous landscapes. In contrast, **LCSA** enables **Gradient Borrowing**: it allows $[\nabla_\nu \mathcal{L}_{cls}]_i \neq 0$ even if the direct gradient is zero, provided a descendant $j$ has a strong gradient ($[\nabla_\delta \mathcal{L}_{cls}]_j \neq 0$) and is causally sensitive to $i$. By aggregating downstream signals via the learned causal structure, LCSA effectively acts as a look-ahead mechanism, guiding the optimizer out of local optima while adhering to the data manifold.

## E. Experimental protocol

**Datasets.** Consistent with previous work(Simonetto et al., 2024a;b), we evaluate our method on five public datasets; Table 8 summarizes their statistics.

- **URL.** This dataset is designed for phishing detection, where malicious URLs mimic legitimate sites to steal personal or financial information. Simonetto et al. (2024a;b) extract a variety of features—such as counts of specific substrings (e.g., "www", "&", ";"), URL length, port number, presence of brand names, as well as features from external services (e.g., Google indexing status, page rank, DNS presence)—for distinguishing phishing from benign URLs. Fourteen relational constraints are extracted, including seven linear constraints (e.g., hostname length $\leq$ total URL length) and seven Boolean constraints (e.g., if the count of "http" $> 0$ then the slash "/" count $> 0$).
- **LCLD.** Each record in this dataset represents an accepted loan, which is either fully repaid or charged off. Following Simonetto et al. (2024a;b), only samples with "loan status" in {Fully paid, Charged Off} are retained (binary classification). Features unavailable at origination or with $> 30\%$ missing values, as well as redundant categorical features, are removed or aggregated; categorical variables are one-hot encoded. The final version contains 27 input features and 1 target variable, with feature boundary constraints set according to training set extrema. Nineteen Lending Club-controlled features are treated as immutable, and ten feature relations (three linear, seven nonlinear) are defined.
- **CTU.** The CTU-13 dataset (Chernikova & Oprea, 2022) contains labeled network traffic from a university, distinguishing between benign and botnet events. Raw network logs are aggregated by port. The data include 143,000 training and 55,000 test samples, with botnet traffic forming just 0.74%. Of 756 features, 432 are mutable. Researchers identify

constraints both on connection counts (to prevent their artificial reduction) and on protocol properties (e.g., TCP/UDP single-packet maximum size of 1500 bytes), resulting in 360 constraints.

- **WIDS.** Proposed by Lee et al. (2020), this dataset contains critical care survival data for ICU patients. The prediction target is patient survival, given biological and clinical features; the data are highly imbalanced. Thirty linear relational constraints are included.
- **MALWARE.** The Malware dataset (Dyrmishi et al., 2023), is a high-dimensional benchmark derived from a collection of benign and malicious Portable Executable (PE) files. Comprising 17,584 samples with 24,222 features, the dataset captures a diverse set of attributes, including DLL and API imports, PE section headers, and byte-level statistical properties. The sheer scale of the feature space, combined with the complex constraints inherent to executable file structures, makes this dataset particularly challenging for adversarial attacks. The primary task is binary classification to distinguish malware from benign software.

**Architectures.** We focus on the most widely used deep learning models for tabular data:

- **TabTransformer** (Huang et al., 2020) leverages self-attention to transform categorical features into contextual embeddings, enhancing both interpretability and robustness to noisy inputs.
- **TabNet** (Arik & Pfister, 2021) employs a multi-step decision process with sequential attention, dynamically selecting features at each stage and aggregating information for prediction.
- **RLN** (Regularization Learning Networks) (Shavitt & Segal, 2018) minimizes counterfactual loss by learning regularization coefficients for network weights, resulting in highly sparse models and reduced sensitivity.
- **STG** (Stochastic Gates) (Yamada et al., 2020) performs neural network feature selection via a stochastic gating mechanism, implementing a probabilistic relaxation of the feature $l_0$-norm.
- **VIME** (Value Imputation for Mask Estimation) (Yoon et al., 2020) combines deep encoders and predictors with self-supervised and semi-supervised learning for feature imputation and completion.

**Baselines.** To comprehensively evaluate the performance of LCSA, we benchmark it against a diverse set of widely adopted adversarial attacks in the tabular domain. Our comparison encompasses both **gradient-based optimization methods** and **heuristic search-based approaches**.

- **PGD** (Madry et al., 2017) serves as the standard white-box benchmark. It employs projected gradient descent to iteratively maximize the loss within a defined $\ell_p$-norm ball. In the tabular context, we adapt the projection step to respect variable-specific constraints (e.g., ranges for continuous features).
- **Adv-VAE** (He et al., 2025c) is a generative manifold attack that leverages a mixed-input Variational Autoencoder (VAE) to construct imperceptible adversarial examples. Unlike gradient-based methods that optimize in the input space, Adv-VAE searches for perturbations within the continuous latent manifold, decoding them back to the original space to ensure that the generated samples remain statistically indistinguishable from the data distribution while respecting feature constraints.
- **CaFA** (Ben-Tov et al., 2024) is a recently proposed framework specifically optimized for constrained adversarial generation. It balances computational efficiency with attack success rate by enforcing strict adherence to logical and feature-specific constraints inherent to tabular data.
- **CAPGD** (Simonetto et al., 2024a) extends the standard PGD by incorporating sophisticated projection operators capable of handling complex constraint sets (e.g., $L_0$ sparsity combined with $L_\infty$ bounds). It represents the state-of-the-art in handling the mixed-integer nature of tabular perturbations via gradient descent.
- **BF\*** (Kulynych et al., 2018) is a heuristic search algorithm based on the Best-First Search strategy. It treats the feature space as a graph and iteratively explores perturbations to identify the shortest path to a successful attack, making it particularly effective for discrete or non-differentiable feature spaces.
- **MOEVA** (Simonetto et al., 2021) leverages a Multi-Objective Evolutionary Algorithm to generate adversarial examples. Unlike gradient methods, MOEVA treats the attack as a black-box optimization problem, simultaneously minimizing perturbation magnitude and maximizing misclassification confidence.

**Hyperparameter Settings.** We strive for a fair comparison by adhering to standard configurations. For BF\*, CAPGD, Adv-VAE, and MOEVA, we adopt the default hyperparameters recommended in their respective original publications to ensure optimal performance. For the gradient-based baselines PGD and CaFA, we standardize the computational budget to 20 iterations. The step size is set to $\alpha = 0.1$, and we employ 5 random restarts to mitigate the risk of getting trapped in local maxima. For our proposed LCSA, we specify the settings for both phases to ensure reproducibility. In **Phase I**, we parameterize each mechanism (Eq. 7) as a 2-layer MLP with 64 hidden units (128 for high-dimensional data) and Leaky

ReLU activations. The ensemble is optimized using Adam (learning rate $5 \times 10^{-4}$) for 50 iterations. To robustly extract the DAG structure from continuous weights $W^{(m)}$, we employ an adaptive thresholding strategy rather than a fixed cutoff. Specifically, for each model, we set the threshold to the $90^{\text{th}}$ percentile of the absolute weight distribution. This ensures we retain only the top $10\%$ most significant structural dependencies, filtering out low-magnitude noise induced by floating-point optimization. In **Phase II**, we set the warm-up period $T_{\text{warm}} = 4$, ensemble number $M = 10$, penalty $\rho = 0.5$, learning rate $\eta = 0.5$ and the total iterations $T_{\text{total}} = 20$. Furthermore, we adopt an adaptive strategy for sparsity cap $k$: for the high-dimensional MALWARE dataset, we set the candidate parent limit to $k = 150$ to ensure computational tractability. For all other datasets ($d \leq 756$), we set $k = d$ (no screening) to preserve full structural connectivity.

**Hardware.** All experiments were conducted on a High-Performance Computing cluster. Each compute node is equipped with two AMD EPYC 7H12 CPUs (a total of 128 cores @ 2.6 GHz) and 256 GB of RAM. For each experimental run, we allocated 32 CPU cores and 64 GB of RAM.

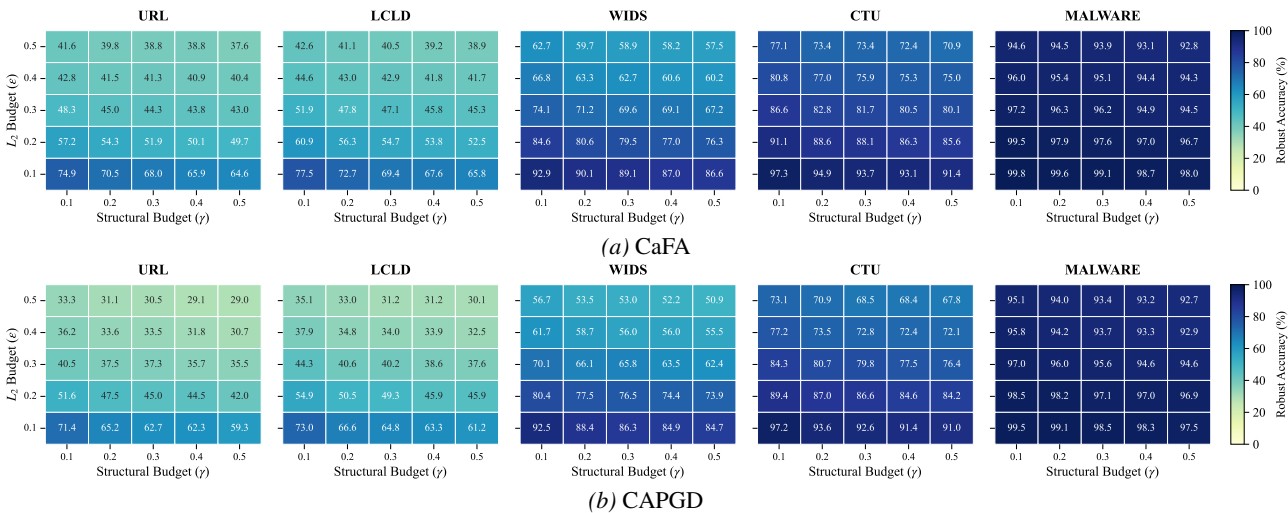

*Figure 5.* Evaluation of robust accuracy under varying $L_2$ and $\gamma$ perturbation budgets on five datasets

## F. Sensitivity Study of Perturbation Budget (*RQ2*)

To benchmark the sensitivity of LCSA, we compare it against the two most competitive baselines identified in Table 2: CAPGD and CaFA. As visualized in Fig. 5, while all methods adhere to the expected trend where robust accuracy increases as perturbation budgets ($L_2$ and $\gamma$) tighten, LCSA consistently uncovers more vulnerable subspaces. On the URL dataset, LCSA demonstrates a substantial lead; under the loosest budget ($L_2 = 0.5, \gamma = 0.5$), LCSA reduces robust accuracy to 10.0%, significantly outperforming CAPGD (29.0%) and CaFA (37.6%). This advantage persists even under stricter constraints. For instance, at the stringent setting of $L_2 = 0.1$ (row 5), LCSA maintains a robust accuracy of roughly 48%–62%, whereas the baselines degrade rapidly to over 70%, failing to identify valid adversarial examples. Most notably, on the resilient MALWARE dataset where baselines struggle to penetrate the model's defenses (yielding $\geq 92\%$ accuracy even at $L_2 = 0.5$), LCSA successfully suppresses the robust accuracy to 89.2%. This empirical evidence confirms that LCSA's structural constraints allow it to navigate the perturbation space more efficiently than other methods, regardless of the geometric or structural budget tightness.

## G. Ablation Study (*RQ7*)

To rigorously evaluate the contribution of each component in LCSA, we conduct a comprehensive ablation study across all five datasets. The results, reported in Table 9, measure both attack effectiveness (Robust Accuracy) and structural plausibility (Validity). We analyze the impact of critical components.

**Impact of Ensemble Size ($M$).** We investigate the necessity of the ensemble strategy by varying the number of SCMs, $M$, from 1 to 10. The results reveal a clear monotonic trend: increasing $M$ consistently improves both attack success and validity. With a single model ($M = 1$), the *Validity* drops significantly (e.g., 78.9% on MALWARE vs. 96.2% for $M = 10$). This suggests that a single SCM fails to capture epistemic uncertainty, leading to overconfident but erroneous structural constraints. While $M = 5$ achieves performance close to our default ($M = 10$), the full ensemble ensures maximum

*Table 9.* Ablation Study on each component in LCSA across five datasets ($L_2 = 0.5, \gamma = 0.5$). **Rob.** ($\downarrow$): Robust Accuracy (lower is better). **Val.** ($\uparrow$): Validity, the percentage of adversarial examples satisfying the structural constraint ($\mathcal{L}_{\mathrm{scm}}(\tilde{x}) - \mathcal{L}_{\mathrm{scm}}(x) \leq \gamma$). The best results are highlighted in **bold**.

| Method Variant | URL | | LCLD | | WIDS | | CTU | | MALWARE | |
|---|---|---|---|---|---|---|---|---|---|---|
| | Rob. | Val. | Rob. | Val. | Rob. | Val. | Rob. | Val. | Rob. | Val. |
| **LCSA** ($M = 10$) | **10.3%** | **98.5%** | **11.7%** | **97.2%** | **38.4%** | **96.8%** | **58.0%** | **99.1%** | **89.6%** | **96.2%** |
| *w/o* Ensemble ($M = 1$) | 14.8% | 81.2% | 16.5% | 78.8% | 43.8% | 82.4% | 63.5% | 85.1% | 91.4% | 78.9% |
| *w/o* Ensemble ($M = 3$) | 12.6% | 86.5% | 14.3% | 85.1% | 41.2% | 88.6% | 61.2% | 90.3% | 90.9% | 84.3% |
| *w/o* Ensemble ($M = 5$) | 11.9% | 96.8% | 12.3% | 95.4% | 39.5% | 93.2% | 59.4% | 97.8% | 89.8% | 95.5% |
| *w/o* Hetero. Mech. | 18.7% | 62.3% | 21.3% | 58.9% | 48.2% | 65.1% | 68.9% | 72.0% | 93.1% | 55.4% |
| *w/o* Soft-Emb. (use STE) | 15.2% | 89.0% | 16.9% | 91.2% | 43.8% | 90.5% | 64.1% | 94.2% | 91.8% | 88.7% |
| *w/o* Soft-Emb. (use GS, $\tau = 0.1$) | 17.8% | 87.5% | 21.3% | 89.4% | 48.5% | 87.2% | 69.8% | 91.5% | 93.2% | 86.4% |
| *w/o* Soft-Emb. (use GS, $\tau = 0.5$) | 12.6% | 92.4% | 18.1% | 93.5% | 45.2% | 92.3% | 67.8% | 95.8% | 90.5% | 91.4% |
| *w/o* Soft-Emb. (use GS, $\tau = 1.0$) | 16.5% | 89.2% | 19.8% | 90.7% | 47.6% | 89.9% | 68.5% | 90.3% | 92.5% | 88.1% |
| *w/o* ALM Optimization | 14.3% | 71.2% | 19.1% | 68.7% | 39.2% | 70.4% | 61.8% | 75.1% | 90.4% | 68.9% |
| *w/o* Warm Start | 10.9% | 93.4% | 12.8% | 92.1% | 42.8% | 91.5% | 60.2% | 95.8% | 90.6% | 90.1% |
| Structure-Blind (PGD + Soft-Emb.) | 23.3% | 64.4% | 31.5% | 55.8% | 66.4% | 74.3% | 93.1% | 73.2% | 94.2% | 70.3% |

stability and validity across diverse datasets.

**Necessity of Heterogeneous Mechanisms.** A core innovation of LCSA is the distinct modeling of continuous and categorical variables. We replace the heterogeneous architecture with a homogeneous Gaussian SCM. In this variant, all variables (including one-hot encoded categorical ones) are treated as continuous variables modeled by standard MSE loss, ignoring the simplex constraint. This variant suffers the most severe performance degradation. On the MALWARE dataset, *Validity* plummets to 55.4%, and Robust Accuracy rises to 93.1% (indicating a failed attack). This confirms that imposing Gaussian assumptions on discrete tabular data creates a fundamental distribution mismatch, generating adversarial examples that are easily rejected by the structural constraint.

**Effectiveness of Soft-Embedding.** Conversely, we evaluate LCSA without soft-embeddings, using standard relaxations (e.g., Straight-Through Estimator **and Gumbel-Softmax**) instead. Robust Accuracy increases across the board (e.g., 15.2% **for STE and 12.6%–17.8% for GS** on URL), indicating that noisy gradients from standard relaxations hinder optimization. However, comparing this to the Structure-Blind row reveals a critical insight: *LCSA with noisy gradients (w/o Soft-Embed)* still outperforms *Structure-Blind PGD with perfect gradients* ($\leq$**17.8%** vs **23.3%** on URL). This definitively proves that structural guidance is the primary driver of performance, with soft-embeddings playing a supportive role.

**Stability of ALM Optimization.** We replace ALM with a standard Penalty Method, which adds a static weighted penalty term $\lambda \cdot \max(0, \mathcal{L}_{\mathrm{scm}} - \gamma)$ to the objective function without dynamically updating Lagrange multipliers. While the Penalty Method can sometimes achieve low Robust Accuracy (e.g., 39.2% on WIDS), it fails to strictly maintain the structural constraint, as evidenced by the sharp drop in *Validity* (e.g., only 68.9% on MALWARE). ALM successfully balances the trade-off, ensuring the attack is successful only when it is strictly structurally valid.

**Role of Warm Start Strategy.** We skip the warm-up stage and strictly enforce the structural constraint from the very first iteration. The results indicate that the warm start is beneficial for navigating the complex constrained landscape. Without it, the optimizer often gets trapped in local optima where the constraints are satisfied, but the attack strength is suboptimal. For instance, on the WIDS dataset, removing the warm start degrades attack performance, increasing Robust Accuracy from 38.1% to 42.8%. This suggests that the initial unconstrained exploration helps identify more vulnerable directions.

**Disentangling Differentiability from Structure.** A key question is whether LCSA's performance gain stems from the *Ripple Structure* or merely from the differentiable *Soft-Embedding* enabling better gradient flow. To isolate these factors, we evaluate a **Structure-Blind (PGD + Soft-Emb.)** variant. This baseline utilizes the same soft-embedding relaxation as LCSA but optimizes independent perturbations without the Ripple mechanism. As shown in Table 9, although Soft-Embedding enables gradient descent, the lack of structural guidance leads to significantly worse performance. On CTU, Robust Accuracy degrades to 93.1% compared to LCSA's 58.0%, indicating that the optimizer fails to find the adversarial subspace. Furthermore, the structural *Validity* drops considerably (e.g., 70.3% on MALWARE vs 96.2% for LCSA). This result confirms that differentiability alone is insufficient; the *Ripple mechanism* is essential for both guiding the optimizer towards effective attacks and ensuring these attacks respect the complex data manifold.

*Table 10.* Average Logical Violation Rate (LVR) across 5 Model Architectures. We report the percentage of adversarial examples that violate explicit domain-specific constraints. Results are averaged across all five target models (TabTr, RLN, VIME, STG, TabNet)

| Dataset | $L_2 = 0.5 \& \gamma = 0.5$ | | | | | | | $L_\infty = 0.5 \& \gamma = 0.5$ | | | | | | |
|---|---|---|---|---|---|---|---|---|---|---|---|---|---|---|
| | PGD | Adv-VAE | CaFA | CAPGD | BF* | MOEVA | **LCSA** | PGD | Adv-VAE | CaFA | CAPGD | BF* | MOEVA | **LCSA** |
| URL | 32.4% | 38.2% | 23.6% | 25.7% | 33.1% | 29.5% | **4.1%** | 35.2% | 41.1% | 31.3% | 26.7% | 36.5% | 32.1% | **6.5%** |
| LCLD | 58.5% | 45.6% | 52.1% | 48.9% | 60.2% | 54.6% | **22.6%** | 62.1% | 54.3% | 55.4% | 52.2% | 64.5% | 58.1% | **28.1%** |
| CTU | 72.1% | 75.4% | 63.2% | 65.7% | 74.5% | 69.2% | **51.5%** | 76.5% | 64.9% | 71.8% | 67.2% | 78.2% | 72.5% | **43.8%** |
| WIDS | 45.6% | 42.9% | 42.3% | 34.6% | 47.1% | 43.5% | **13.1%** | 48.3% | 34.6% | 45.9% | 41.9% | 50.2% | 46.8% | **19.4%** |
| MALWARE | 25.2% | 32.8% | 22.1% | 18.7% | 26.1% | 23.4% | **7.5%** | 29.7% | 30.2% | 25.4% | 20.5% | 29.5% | 26.1% | **13.9%** |

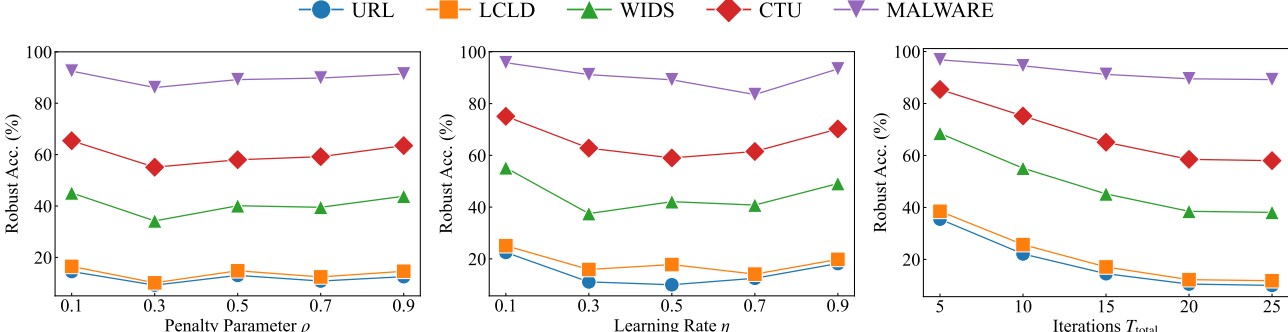

*Figure 6.* Hyperparameter sensitivity of penalty parameter $\rho$, learning rate $\eta$, and iterations $T_{\text{total}}$ across five datasets

## H. Hyperparameter Study (*RQ7*)

We evaluate the impact of the penalty parameter $\rho$, learning rate $\eta$, and iterations $T_{\text{total}}$ on attack performance. As shown in Fig. 6, our analysis reveals three key observations: (i) **Penalty Parameter** $\rho$: Performance follows a convex trend, peaking at $\rho = 0.5$. Lower values ($\rho < 0.3$) fail to enforce valid constraints, while excessive penalties ($\rho > 0.7$) dominate the loss, hindering the maximization of classification error. (ii) **Learning Rate** $\eta$: A Fig. 6 indicates that $\eta = 0.5$ is optimal. Conservative rates ($\eta = 0.1$) lead to slow convergence, whereas aggressive rates ($\eta = 0.9$) cause oscillation, preventing precise localization of the adversarial subspace. (iii) **Iterations** $T_{\text{total}}$: The robust accuracy decreases rapidly and stabilizes around 20 iterations. We adopt $T_{\text{total}} = 20$ as a default, confirming LCSA's efficiency in identifying high-quality structural perturbations with minimal computational cost.

## I. Structural Plausibility on Real-World Data

To rigorously evaluate mechanistic plausibility on real-world data without ground-truth structural equations, we introduce the **Logical Violation Rate (LVR)**. Formally, let $\mathcal{D} = \{x_1, \ldots, x_N\}$ be the set of benign samples and $\mathcal{C} = \{c_1, \ldots, c_K\}$ be the set of explicit domain-specific logical constraints. Let $\tilde{x}_i$ denote the raw adversarial example generated for $x_i$ prior to any validity projection. The LVR is defined:

$$\text{LVR} = \frac{1}{N} \sum_{i=1}^{N} \mathbb{I}\left(\exists k \in \{1, \ldots, K\} \text{ s.t. } c_k(\tilde{x}_i) \text{ is False}\right) \times 100\%, \tag{26}$$

where $\mathbb{I}(\cdot)$ is the indicator function. As presented in Table 10, gradient-based baselines (e.g., PGD, CAPGD) and heuristic searches (BF*, MOEVA) exhibit consistently high violation rates, peaking at over 60% on constrained datasets like LCLD and CTU. This confirms that these algorithms minimize classification loss by exploiting logical loopholes—feature directions that are mathematically optimal but semantically impossible. While the generative Adv-VAE attempts to constrain perturbations via a latent manifold, its performance remains inconsistent (e.g., higher violations than PGD on URL), suggesting that its symmetric Gaussian assumptions struggle to capture the sharp, asymmetric logic of tabular data. In sharp contrast, **LCSA consistently achieves the lowest violation rates across all settings**, reducing structural errors by a factor of **2× to 5×** compared to strong baselines (e.g., dropping LVR from 49% to 22.6% on LCLD). Although the non-zero violations reflect the inherent challenge of approximating discrete logic with continuous SCMs, this substantial reduction empirically validates that the **Ripple Mechanism** successfully internalizes structural dependencies, guiding the optimization toward structurally valid regions where other methods fail.

