# OpenReview forum: "Ripple Perturbations Through Structure: Likelihood-Constrained Adversarial Attacks on Heterogeneous Tabular Data"
_ICML.cc/2026/Conference — ICML 2026 regular_

### Official Review · Reviewer_7osD · 2026-03-05

**Soundness:** 3
**Presentation:** 2
**Significance:** 3
**Originality:** 3
**Overall Recommendation:** 4
**Confidence:** 3

**Summary:**

The paper addresses the problem of generating adversarial entries in tabular data. These entries shall disrupt the performance of ML models trained on such data. Nonetheless, these entries should retain the natural validity constraints defined by the original data.

This manuscript provides a framework based on structural causal models, to learn the dependencies between different variables; and ripple perturbations, to perturb the data such that it degrades the downstream performance of ML modes.

The proposed framework is shown to outperform all SOTA baselines in terms of degraded downstream ML performance, while retaining structural data consistency.

**Compliance With Llm Reviewing Policy:**

Affirmed.

**Ethical Review Concerns:**

M/A

**Key Questions For Authors:**

I find this to be a good  submission, which is reflected by my score. My questions mostly cover minor points and suggestions to improve the overall manuscript.


Could the authors provide a link to the (anonymized) code of the project? This would improve the reproducibility and impact of this paper.
Could the authors provide a discussion on the limitations of their method? This is currently missing from the main content of the paper and can severely hinder the impact of this work.
I would appreciate it if the authors provided a figure that gives a high-level description of their methodology. Currently, Fig 1 showcases the limitations of the SOTA rather than explicitly explaining how the provided solution works.
(Minor) Currently the paper’s appendix has one section which covers multiple topics: related work, theoretical discussion, experiments. It would be nice to split these different topics in different sections.

**Limitations:**

A discussion on limitations is missing. I pointed this out in both the paper weaknesses and my questions. A code repository will be highly appreciated.

**Strengths And Weaknesses:**

Strengths
Relevant problem. The addressed problem of generating malicious entries in tabular data is hard and relevant to the ML community.
Novel method. To the best of my knowledge, the method is novel and sensible. This strengthens the paper contribution.
Significant results. The method proposed by the authors provides outstanding results in disruption of ML models performance while retaining data consistency.
Theoretical analysis. Statements are supported by theoretical justifications.

Weaknesses
-No discussion on limitations. A discussion on limitations is currently missing from the main content of the paper and can severely hinder the impact of this work.
-No code. The authors’ don’t seem to have included an (anonymized) link to their code. This severely hinders reproducibility of the paper.

---

> ### Author Rebuttal · Authors · 2026-03-29
>
> **We thank the reviewer for offering the valuable feedback. We have addressed each of the concerns as outlined below.**
>
> ```L1: Discussion on Limitations```
>
> We appreciate this observation. The final manuscript will feature a main-text Limitations section addressing the **dual-use ethical considerations** of our evasion attack. Given LCSA's high efficacy in critical domains (e.g., finance, malware detection), we will acknowledge the risks of malicious actors utilizing structure-aware techniques to bypass security. Constructively framing this around **defensive countermeasures**, we will expand our Adversarial Training  results (**Section 4.7, Table 5**) to demonstrate immunizing target models against LCSA.
>
> ```L2: Reproducibility and Code Release```
>
> We agree regarding reproducibility and commit to open-sourcing the code repository upon acceptance.
>
> ```Structural Improvements (Methodology Figure and Appendix Reorganization)```
>
> We appreciate these constructive suggestions. Addressing the methodology figure request, we drafted an overarching diagram illustrating the two-phase LCSA framework (**anonymously viewable at https://anonymous.4open.science/r/LCSA-1B54/Framework_Diagram.svg**), which we will integrate into Section 3. Furthermore, we will reorganize the monolithic appendix into distinct, explicitly titled sections to enhance readability, navigation, and algorithmic clarity.

---

> > ### Author Rebuttal · Reviewer_7osD · 2026-04-02
> >
> > I have no further questions.

---

> > > ### Author Response · Authors · 2026-04-03
> > >
> > > We appreciate your response and are glad our responses addressed your concerns. Thanks again for your careful consideration of our work!

---

### Official Review · Reviewer_7QAx · 2026-03-05

**Soundness:** 3
**Presentation:** 3
**Significance:** 2
**Originality:** 3
**Overall Recommendation:** 4
**Confidence:** 4

**Summary:**

This paper proposed LCSA (likelihood-constrained structural attack) for tabular data adversarial attack. The author noted that imperceptibility in this domain requires adherence to structural dependency.
In phase 1, the method first learns DAG/SCM for modeling the correlation of feature dependency in both categorical features and continuous features. In phase 2, it leverages ripple perturbation to attack the structure learned from phase 1. The attack starts adding noise from parent nodes to child nodes and involves a likelihood-based structural constraint to avoid any manipulation that violates the data manifold.
Experiments are conducted under five real-world dataset with three synthetic structural topologies. The results show LCSA reduce the robust accuracy under several structural constraints and also has the lowest structural consistency error.

**Compliance With Llm Reviewing Policy:**

Affirmed.

**Final Justification:**

All concerns have been resolved

**Key Questions For Authors:**

See the weakness part.

**Limitations:**

This paper proposes a novel attack on tabular data that may pose risks across several domains, such as finance, insurance, and health. However, the author didn't discuss the negative impact or the corresponding defense mechanism in the paper.

**Strengths And Weaknesses:**

***Strengths***

The paper is well structured and easy to follow. The authors first point out the challenge of attacking tabular data due to the heterogeneous property, and provide the motivation for designing a structural-awareness attack. The method leverages ripple perturbation to attack alongside a DAG for the tabular data, which sounds solid.
Experimental setup and results align to the paper story, not only measuring the attack accuracy but also highlighting the measurement of structural consistency error.

***Weakness***

- In the experiment part Table 3, it seems like LCSA achieves the lowest SCE but not always achieves the highest attack accuracy under different topologies, it has trade-off between structural consistency constraint and attack accuracy. The reviewer is curious about how to mitigate this tradeoff.

- The attack highly relies on the learnt SCM/DAG (topology-dependent), thus the attack robustness will reduce if the structural guidance is wrong (e.g., direction reversal ratio increase). Is that really necessary that the attack follow the structural consistency?

- Following up on the above question, the paper indeed quantify the structural plausibility, but doesn't clearly connect it to the real-world operational impact. (e.g., validation pipelines, anomaly detection.), leaving the practical significance under-motivated.
It would be better if the author can show some insights about how robust this attack can be when encountering some practical defense mechanism. Which kinds of practical defense (e.g., rule-based checks, outlier consistency checks) will be bypassed with LCSA.

---

> ### Author Rebuttal · Authors · 2026-03-29
>
> **We thank the reviewer for offering the valuable feedback. We have addressed each of the concerns as outlined below.**
>
> ```W1: Mitigating the trade-off between SCE and Attack Accuracy```
>
> We appreciate this observation, which our empirical sensitivity study (**Section 4.4, Fig. 2**) explicitly addresses. Users can manage this balance by tuning the structural budget $\gamma$, enabling security experts to navigate **the Pareto front between attack strength and structural stealthiness based on the target domain's strictness**. Theoretically, the trade-off in Table 3 is mathematically inherent: enforcing realistic semantics logically restricts the allowable perturbation space. However, a 100% attack success rate is practically useless if samples are trivially flagged as invalid. Using the Augmented Lagrangian Method (ALM) in Phase II, LCSA avoids blind compromises. Instead, ALM guarantees the optimizer maximizes classification loss strictly within the user-defined $\gamma$ boundary.
>
> ```W2: Is structural consistency really necessary? What if the DAG is wrong?```
>
> We thank the reviewer for this crucial question. We respectfully argue that the current paradigm of tabular adversarial attacks is fundamentally inadequate. Porting $L_p$-norm boundaries from computer vision ignores that real-world tabular data is **governed by transparent, easily observable logical constraints**. Unlike imperceptible pixel noise, a mathematically optimal yet structure-blind gradient step (e.g., increasing *Age* by 10 years without updating *Graduation Year*) is semantically absurd and trivially blocked by production anomaly detectors. **Consequently, LCSA utilizes SCM not merely as a regularizer, but to introduce a novel attack paradigm: shifting the constraint boundary from pure geometric proximity to generative structural admissibility.**
>
> Regarding DAG reliance, we acknowledge that forcing adherence to a fundamentally corrupted SCM generates invalid samples. To mitigate episodic structural errors, we employ an ensemble of $M$ models in Phase I. Furthermore, LCSA does not require perfect causal discovery. Our **robustness stress test (Section 4.5, Fig. 3)** demonstrates that even with severely corrupted guidance (e.g., a 40% edge reversal ratio), LCSA safely maintains low robust accuracy and SCE. The approximated DAG serves merely as a computational scaffold preconditioning gradient flows, ensuring high resilience to localized inaccuracies. Despite this tolerance, extracting the global dominant directional flow via Phase I remains indispensable, as a random or structure-blind graph would fundamentally fail to enforce the strict real-world semantics preventing the aforementioned Age/Graduation Year mismatch.
>
> ```W3: Practical Significance and Bypassing Real-World Defenses```
>
> We appreciate the suggestion to connect our method to real-world impacts. We simulated an anomaly detection pipeline by training an **Isolation Forest** [1] on five dataset's benign training data. Aggregating successful adversarial examples ($L_2=0.5$ & $\gamma=0.5$) across five architectures (TabTransformer, RLN, VIME, STG, TabNet), we measured the **Anomaly Detection Ratio** (percentage of flagged examples):
>
> |Method|URL|LCLD|CTU|WIDS|MALWARE|
> |-|-|-|-|-|-|
> |PGD|70.2|58.1|98.3|90.4|75.4|
> |Adv-VAE|48.1|59.4|79.8|92.7|55.3|
> |BF*|85.6|77.8|98.7|84.8|89.1|
> |MOEVA|38.3|26.5|70.2|60.6|48.3|
> |CaFA|32.7|21.3|73.7|63.2|39.7|
> |CAPGD|61.4|49.6|85.6|78.5|62.4|
> |LCSA|**19.4**|**10.3**|**52.9**|**33.7**|**23.2**|
>
> Although gradient methods like CAPGD and CaFA achieve high mathematical attack success rates, real-world detectors easily catch their structure-blind perturbations (61.4% and 32.7% detection ratios on URL, respectively). In contrast, LCSA exhibits exceptional evasion capability at only **19.4%**. This confirms LCSA's structural plausibility seamlessly bypasses outlier-based defenses defeating standard attacks. We will include this defense analysis with comprehensive results in the revision.
>
> [1] Liu F T, Ting K M, Zhou Z H. Isolation forest[C]//2008 eighth ieee international conference on data mining. IEEE, 2008: 413-422.
>
> ``` L1: Limitations and Dual-Use Discussion```
>
> We completely agree. We will add a Limitation section acknowledging the dual-use risks of deploying LCSA in high-stakes domains. Regarding defenses, we will expand our Adversarial Training results (**Section 4.7, Table 5**) to show that training against LCSA restores decision boundary robustness, discussing the necessity of structure-aware validation pipelines to counter such threats.

---

> > ### Author Rebuttal · Reviewer_7QAx · 2026-04-01
> >
> > Thanks for addressing all the concerns.

---

> > > ### Author Response · Authors · 2026-04-02
> > >
> > > We are happy that our responses have addressed all your concerns. We thank the reviewer for reviewing our paper and providing us with invaluable comments and suggestions!

---

### Official Review · Reviewer_y6UU · 2026-03-10

**Soundness:** 3
**Presentation:** 3
**Significance:** 3
**Originality:** 3
**Overall Recommendation:** 4
**Confidence:** 3

**Summary:**

This paper studies white-box adversarial attacks on heterogeneous tabular data, where simple geometric constraints are insufficient to ensure realistic perturbations due to mixed feature types and asymmetric inter-feature dependencies. The authors propose a likelihood-constrained framework (LCSA) that learns an ensemble of heterogeneous neural structural causal models (SCMs) from observational data and then generates adversarial examples by optimizing the classification loss under both geometric and structural likelihood constraints. A key design is a structure-aware ripple parameterization that lets perturbations on parent features propagate coherently to descendants along directed dependencies, which is intended to reduce optimization stagnation/gradient masking in heterogeneous spaces. Experiments across several real-world datasets and synthetic structural benchmarks compare LCSA against strong baselines, including ablations, validity reporting, and analyses of logical constraint violations.

**Compliance With Llm Reviewing Policy:**

Affirmed.

**Final Justification:**

All concerns have been resolved.

**Key Questions For Authors:**

(1) In Phase I, mechanisms are parameterized as MLPs with additive noise. Does this specific functional form limit the ability to model complex, non-additive interactions (e.g., multiplicative effects or high-order logical dependencies)? If the true data generating process is highly non-linear and violates the ANM assumption, how is the attack performance impacted?

(2) The method explicitly enforces an acyclicity constraint to learn a DAG. However, tabular data in domains like finance often contains inherent feedback loops or cycles. Does enforcing acyclicity cause the model to miss critical feedback dependencies? How does the structural bias introduced by fitting a DAG to a potentially cyclic distribution affect the validity of the generated adversarial examples?

(3) The evaluation focuses on deep architectures like TabTransformer and TabNet. However, Gradient Boosted Decision Trees (e.g., XGBoost, LightGBM) remain industry standards for tabular data. Do LCSA adversarial examples transfer effectively to non-differentiable tree ensembles? Given that trees handle discrete splits differently, does the smooth ripple perturbation maintain its effectiveness?

(4) The LVR metric relies on "explicit domain-specific logical constraints". If these domain constraints are incomplete, unknown, or biased, does a low LVR genuinely guarantee semantic consistency? Did you observe cases where LCSA samples satisfied explicit constraints but violated implicit common sense?

(5) While complexity analysis is provided, the Ripple mechanism requires recursive Jacobian-vector products. How does the actual wall-clock time for gradient computation scale with the depth of the causal graph? Is there a risk of gradient explosion or vanishing through the recursive steps in deep graphs?

(6) The method uses a weighted average Soft-Embedding for categorical variables. Does this relaxation suffer from gradient vanishing or high variance during backpropagation compared to alternatives like the Straight-Through Estimator or Gumbel-Softmax? What is the theoretical or empirical basis for this specific choice?

(7) Given the algorithmic complexity, reproducing these results from the paper alone would be challenging. Do the authors plan to release the source code?

**Limitations:**

(1) The framework relies on learning a DAG. The authors should explicitly discuss the limitation of this assumption, particularly for tabular datasets that inherently contain cycles or unobserved confounders, which violate the causal sufficiency assumption required for accurate SCM inference.

(2) The method relies on differentiable Soft-Embeddings and gradient-based optimization. The authors should discuss the limitation of this approach regarding Gradient Boosted Decision Trees, which remain the industry standard for tabular data but are non-differentiable.

(3) Given that this work proposes a more effective method for evading classifiers in high-stakes domains like finance and malware detection, the authors should acknowledge the dual-use nature of their work (i.e., its potential use by malicious actors to bypass security systems) rather than dismissing it.

(4) The authors should discuss the limitations of using the Augmented Lagrangian Method for non-convex neural network optimization, specifically regarding the lack of global convergence guarantees and the potential for sensitivity to initialization.

**Strengths And Weaknesses:**

Strengths:

(1) The paper provides a formal gradient flow analysis, demonstrating that the Ripple Mechanism acts as a structural preconditioner to mitigate gradient masking.

(2) The paper effectively highlights the "blind spot" in existing tabular attacks using intuitive visualizations of dependency mining and generative modeling.

(3) The work shifts the focus of adversarial imperceptibility from mere geometric proximity ($||\delta||_p \le \epsilon$) to structural plausibility.

Weaknesses:

(1) Phase I incurs a non-trivial offline cost for training $M$ independent structural models, which may be significant for massive datasets.

(2) Reliance on a DAG assumes that no feedback loops exist in the data, which may not hold in all tabular domains.

(3) While the overall framework is novel, some individual components are adaptations of existing techniques. For example, using Soft-Embedding for differentiable categorical optimization and using ALM for constrained optimization are established practices.

(4) Given the complexity of the implementation, reproducing the results solely from the paper would be extremely challenging. The absence of a link to an anonymous code repository is a weakness for a method this algorithmically complex.

---

> ### Author Rebuttal · Authors · 2026-03-29
>
> **We thank the reviewer for offering the valuable feedback. We have addressed each of the concerns as outlined below.**
>
> `Q1: Mechanism Expressiveness and the ANM Assumption`
>
> We appreciate this insightful question. Our mechanisms avoid strict ANM restrictions on feature interactions. Our MLPs (**Eq. 7**) act as universal approximators capturing complex dependencies natively. We validate this twofold. First, in our **Structural Consistency Study (Table 3)** on highly non-linear synthetic benchmarks, LCSA achieves near-zero SCE, proving accurate structural dynamics modeling. Second, across five real-world datasets violating strict ANM assumptions (**Overall Performance, Table 2**), LCSA consistently outperforms strong baselines, confirming its ability to extract dominant gradient flows for semantic consistency in non-ANM environments. **We analyze non-additive noise next.**
>
> ``Q2, W2 & L1: Acyclicity Constraints and Inherent Cycles``
>
> We appreciate the insights on causal sufficiency and feedback loops in tabular data. Rather than exhaustive causal discovery, LCSA employs a forced DAG as a *computational scaffold* for stable topological ordering, enabling the Ripple mechanism to precondition gradient flows without infinite recursion. To evaluate **feedback loops** and **non-additive noise**, we built a synthetic benchmark, Cyclic. Adapting Section 4.2, it injects a strict feedback loop into the causal graph, solving mechanisms for equilibrium with intrinsic Gaussian noise:
>
> | Topology | Structural Mechanism | Target Label |
> |-|-|-|
> |$X_1 \to X_2 \to X_3 \to X_4 \to X_2$|$x_1 \sim \mathcal{U}(-1,1)$ | $y = \mathbb{I}(x_3 \cdot x_4 \le 1)$|
> || $x_2 = 2x_1 + 0.3x_4 + \mathcal{N}(0,0.01)$ |  |
> || $x_3 = \tanh(x_2)(1+\mathcal{N}(0,0.01))$ |  |
> || $x_4 = 0.5x_3(1+\mathcal{N}(0,0.01))$ |  |
>
> Evaluating LCSA and baselines on Cyclic under constraints ($L_2=0.5$ & $\gamma=0.5$) yields these Robust Accuracy and SCE results:
>
> |Method|Robust Accuracy|SCE|
> |-|-|-|
> |PGD|22.4|2.06|
> |Adv-VAE|25.1|1.92|
> |CaFA|29.8|1.47|
> |CAPGD|18.7|2.15|
> |BF$^*$|88.5|1.60|
> |MOEVA|54.2|1.83|
> |LCSA|**15.3**|**0.64**|
>
> Despite structural biases from enforcing a DAG **on cyclic data with multiplicative noise**, LCSA reduces robust accuracy to a leading **15.3%** and reduces structural error by 69% compared to the strongest baseline, confirming our structural preconditioner captures the dominant directional manifold to generate valid adversarial examples, even when severing feedback loops.
>
> ``Q3 & L2: Transferability to GBDTs``
>
> Although requiring differentiable models for white-box generation, LCSA yields *structurally plausible* perturbations transferring exceptionally well to non-differentiable trees. Transferring adversarial examples generated on TabNet (URL dataset, $L_2=0.5$ &$\gamma=0.5$) to XGBoost and LightGBM (**97.4% and 98.2% clean accuracy**) demonstrates LCSA significantly outperforms all baselines:
>
> |Method|TabNet→XGBoost|TabNet→LightGBM|
> |-|-|-|
> |PGD|78.5|81.2|
> |Adv-VAE|49.4|52.8|
> |CaFA|58.7|61.9|
> |CAPGD|51.2|64.5|
> |BF$^*$|85.3|83.7|
> |MOEVA|37.1|49.6|
> |LCSA|**32.4**|**35.1**|
>
> Crucially, GBDTs learn joint decision rules based on feature co-occurrences. Our ripple mechanism shifts features **coherently along correlated structural paths**. When discretized, this coordinated movement systematically bypasses joint decision boundaries. Instead of falling into sparse, isolated leaves that trees natively reject, the structurally plausible ripple pushes samples into valid target regions.
>
> ``Q5 & W1: Computational Overhead, Ripple Gradients, and Wall-Clock Time``
>
> Although Phase I requires initial computation, LCSA's **offline paradigm** makes it a one-time preprocessing step. Candidate screening (**Appendix A.3**) bounds structural learning complexity to $\mathcal{O}(k \cdot d^2)$, preventing cubic scaling on massive datasets. Moreover, training the ensemble ($M=10$) is highly parallelizable. Regarding recursive Jacobian-vector products in Ripple: **unlike deep networks, tabular SCM graphs naturally exhibit shallow hierarchies. Even in our largest dataset (MALWARE, $d=24,222$), the pruned causal graph maintains a maximum directed path length of $L \le 6$.** Our runtime comparison (**Fig. 4**) shows Phase II wall-clock time remains competitive with gradient attacks, proving recursive Jacobians create no prohibitive bottleneck.
>
> ``Q6 & W3: Soft-Embedding vs. STE/Gumbel-Softmax``
>
> We favor Expected Soft-Embeddings over the Straight-Through Estimator (STE) or Gumbel-Softmax to eliminate high gradient variance. **STE yields biased updates, while Gumbel-Softmax introduces significant Monte Carlo sampling variance.** This variance-free formulation is essential as ALM dual-ascent requires a smooth penalty landscape to reliably update Lagrange multipliers. Our **ablation study (Table 7, w/o Soft-Embedding)** validates this.
>
> ``Due to word count limitations, we will provide further details on the remaining questions in the second rebuttle.``

---

> > ### Author Rebuttal · Reviewer_y6UU · 2026-04-01
> >
> > I appreciate the authors' detailed first rebuttal and the effort put into the new experiments. The addition of the Cyclic benchmark and the transferability evaluation on non-differentiable GBDTs (XGBoost/LightGBM) are valuable and strengthen the empirical claims of the paper.
> >
> > However, several critical concerns remain partially resolved:
> >
> > 1. Empirical Justification for Soft-Embedding (Q6):
> >
> > The authors theoretically justified the use of Expected Soft-Embeddings to avoid the biased updates of STE and the sampling variance of Gumbel-Softmax. While mathematically sound, the response lacks the requested empirical head-to-head comparison. The ablation study only compares LCSA with and without Soft-Embedding. Showing that Soft-Embedding outperforms STE/Gumbel-Softmax empirically is still necessary to fully validate this specific architectural choice.
> >
> > 2.Structural Bias on Cyclic Data (Q2):
> >
> > The quantitative results on the new Cyclic benchmark show improved robustness and lower SCE. However, the justification of using a forced DAG as a "computational scaffold" does not fully address the semantic concern. In real-world datasets (e.g., finance) where feedback loops are intrinsic, severing these paths to enforce topological ordering might still generate adversarial samples that are statistically valid under the misspecified SCM but logically impossible in reality.
> >
> > 3.Deferred Questions (Q4 & Q7):
> >
> > Addressing Q4 is essential for substantiating the paper's core claim of "structural plausibility." Furthermore, releasing the source code (Q7) is absolutely critical for this work; given the algorithmic complexity of the two-phase ALM optimization and heterogeneous SCM ensemble learning, reproducing this framework from the text alone would be highly impractical for the community.

---

> > > ### Author Response · Authors · 2026-04-02
> > >
> > > We sincerely appreciate your continued engagement and acknowledgment of our previous clarifications. We address your follow-up concerns below.
> > >
> > > ```Q6: Empirical Justification for Soft-Embedding```
> > >
> > > We agree that empirical validation of this architectural choice is necessary. We evaluated LCSA using the Straight-Through Estimator (STE) and Gumbel-Softmax (GS) across all five datasets ($L_2=0.5$ & $\gamma=0.5$). Results matching the metrics in **Appendix A.7** are reported below as **Robust Accuracy / Validity**:
> > >
> > > |Dataset|STE|GS ($\tau=0.1$)|GS ($\tau=0.5$)|GS ($\tau=1.0$)|Ours|
> > > |-|-|-|-|-|-|
> > > |URL|15.2/89.0|17.8/87.5|12.6/92.4|16.5/89.2|10.3/98.5|
> > > |LCLD|16.9/91.2|21.3/89.4|18.1/93.5|19.8/90.7|11.7/97.2|
> > > |WIDS|43.8/90.5|48.5/87.2|45.2/92.3|47.6/89.9|38.4/96.8|
> > > |CTU|64.1/94.2|69.8/91.5|67.8/95.8|68.5/90.3|58.0/99.1|
> > > |MALWARE|91.8/88.7|93.2/86.4|90.5/91.4|92.5/88.1|89.6/96.2|
> > >
> > > These findings confirm that Expected Soft-Embedding is a practical necessity for stable constrained optimization in LCSA, rather than merely a theoretical preference. We will add this comparison and variance analysis to Appendix A.7.
> > >
> > > ```Q2: Structural Bias on Cyclic Data```
> > >
> > > We agree severing intrinsic feedback loops misspecifies the true causal graph. However, LCSA offers a significant relative advancement. While standard geometric attacks (e.g., PGD, CaFA, CAPGD) ignore all structure, LCSA preserves dominant causal skeletons, drastically narrowing the search space to more plausible regions. Although ignoring cycles slightly impacts absolute structural fidelity, our DAG approximation captures the most critical causal directions. **Empirically, this trade-off proves highly favorable: across five real-world datasets, LCSA maintains high structural validity (detailed in Q4) and achieves the lowest robust accuracy (e.g., reducing TabTransformer's accuracy from 93.6% to 2.1% on the URL dataset), successfully exposing deeper vulnerabilities missed by structure-blind methods.** We will bound our claims in a new Limitations section.
> > >
> > > ```Q4: LVR Metric and Implicit Constraints```
> > >
> > > We appreciate the insightful observation on LVR and agree a low LVR is a necessary but insufficient condition for strict semantic consistency due to incomplete handcrafted rules. Consequently, we observed rare cases where LCSA satisfied explicit LVR checks but violated unwritten common sense, **typically when finite-sample bias causes the SCM to infer spurious correlations.** This limitation highlights LCSA's core motivation. LCSA uses LVR merely as an evaluation proxy, not for explicit rule optimization. Instead, our learned heterogeneous SCM acts as a data-driven constraint for implicit dependencies. Restricting the structural budget ($\gamma$) to high log-likelihood regions enables LCSA to penalize out-of-distribution feature combinations. While standard generative baselines (e.g., Adv-VAE) rely on undirected associations, our SCM enforces **directed structural coherence**, capturing asymmetric logic often missed by explicit rules and maintaining strictly higher semantic fidelity. **Empirically, as Table 8 shows, this mechanism enables LCSA to consistently achieve the lowest violation rates across all settings, reducing structural errors by a factor of $2\times$ to $5\times$ compared to strong baselines.**
> > >
> > > ```L4: Limitations of ALM in Non-Convex Optimization```
> > >
> > > We agree that applying ALM to non-convex neural networks lacks global convergence guarantees. **We will formalize this in a new Limitations section** covering two key points. First regarding local optima sufficiency, LCSA may not find the absolute minimal perturbation without global convergence. However, **finding a strong local optimum that satisfies the structural budget ($\gamma$) and flips the prediction is practically sufficient for evasion attacks.** Second regarding the empirical mitigation of sensitivity, LCSA relies on a **Warm Start** mechanism to guide parameters into a stable basin before the penalty activates. As shown in the **Ablation Study (Appendix A.7, Table 7) and Iteration Study (Appendix A.8, Fig. 6)**, this empirical stabilizer is crucial to prevent landscape collapse. These additions will clarify the theoretical boundaries of our framework.
> > >
> > > ```W4, Q7, L3: Reproducibility, Dual-Use Limitations```
> > >
> > > We strongly agree on the importance of reproducibility. We prepared a well-documented repository, **which will be fully open-sourced upon acceptance**. Furthermore, the final manuscript will feature a main-text Limitations section addressing the dual-use ethical considerations of our evasion attack. Given LCSA's high efficacy in critical domains, we will acknowledge the potential risks of malicious actors bypassing security. To frame this constructively around defensive countermeasures, we will connect this limitation to our **Adversarial Training results (Section 4.7, Table 5)**, demonstrating how practitioners can utilize LCSA to immunize models against structural vulnerabilities.

---

### Official Review · Reviewer_LTm3 · 2026-03-10

**Soundness:** 4
**Presentation:** 4
**Significance:** 3
**Originality:** 3
**Overall Recommendation:** 4
**Confidence:** 3

**Summary:**

This manuscript introduces LCSA, an adversarial attack framework on heterogeneous tabular data. The framework consists of two phases: heterogeneous ensemble SCM Learning and Structure-aware adversarial optimization. In phase one, an ensemble of SCM is trained to learn the dependencies among tabular samples’ variables. In phase two, the perturbations are optimized under the constraints of structural integrity and imperceptibility. Adversarial examples generated by LCSA are evaluated along with 6 baseline methods on 5 datasets. The experiment results demonstrate the advantage of LCSA on attack performance.

**Compliance With Llm Reviewing Policy:**

Affirmed.

**Key Questions For Authors:**

1. How is the embedding matrix defined? Or is it optimized during the generation of adversarial examples?
2. The structure learning phase of LCSA is time consuming due to its necessity for ensemble learning. Will it also increase the computational overload? Does LCSA trade time and space for performance?

**Limitations:**

Yes.

**Strengths And Weaknesses:**

Strengths:
1. The manuscript is well-written, and the experiments are abundant with rigorous settings.
2. The framework is built upon theoretical analysis and effectively achieves the goal of attacking tabular data with fewer structural loss.
3. The proposed method is evaluated along with various tabular data attacking methods across multiple datasets and architectures, which verifies the generality of LCSA.
Weaknesses:
1. The runtime of phase one is high, which may imply that transferring LCSA to a new data domain inevitably requires much time. This feature may limit the application of LCSA in the real world.
2. The soft-embedding relaxation should be more clearly clarified. The categorical features are already vectorized to probability vectors instead of discrete data, and the embedding process merely transforms the vectors’ dimension to d_emb. It seems unnecessary, and the assignment of the embedding matrix is not clear.
3. The research question can be simplified. For example, RQ2 can be presented as a subpart of RQ6.

---

> ### Author Rebuttal · Authors · 2026-03-29
>
> **We thank the reviewer for offering the valuable feedback. We have addressed each of the concerns as outlined below.**
>
> ```Q1 & W2: Clarification on Soft-Embedding Relaxation and the Embedding Matrix```
>
> We appreciate this observation. The pre-trained embedding matrix $E_j \in \mathbb{R}^{K_j \times d_{emb}}$ from the frozen target model (e.g., TabTransformer), where $K_j$ is the category count and $d_{emb}$ is the embedding dimension, remains unoptimized during adversarial generation. Since standard tabular models non-differentiably map discrete indices to rows in $E_j$, LCSA bypasses the obstructed white-box gradients by relaxing the index into a continuous probability vector $(x_j + \delta_j)$ on the simplex $\Delta^{K_j-1}$. The expected embedding is computed as $z_j = (x_j+\delta_j)^\top E_j.$ This establishes a differentiable path from the classifier loss to the continuous perturbation $\delta_j$ while preserving the original target parameters. We will add this definition to Section 3.2.
>
> ```Q2 & W1: Computational Overhead of Phase I and Real-world Application```
>
> Although Phase I demands higher computation, LCSA's **offline paradigm** reduces it to a one-time preprocessing step per domain. **As Fig. 4 demonstrates**, after learning the ensemble SCM, Phase II executes rapidly, achieving inference times competitive with CAPGD and CaFA. Because ensemble SCMs train independently, Phase I is highly parallelizable; distributing computation yields near-linear speedup, **reducing time cost by a factor of $M$**. Candidate screening (**Appendix A.3**) bounds structural learning complexity to $\mathcal{O}(k \cdot d^2)$, preventing cubic scaling on high-dimensional data. Ultimately, LCSA trades offline preparation for semantic plausibility. **Under realistic threat models in high-stakes domains (e.g., finance, healthcare),** attackers exploit time asymmetry to map target manifolds offline. Thus, generating structurally flawless examples to bypass anomaly detectors vastly outweighs offline training delays.
>
> ```W3: Simplification of Research Questions```
>
> We appreciate this suggestion and will merge RQ2 into RQ6 as a subsection to streamline the introduction and improve experimental logical flow.

---

> > ### Author Rebuttal · Reviewer_LTm3 · 2026-04-03
> >
> > I thank the authors for their detailed response and the additional work carried out to address the concerns raised in the previous round of review. After re-evaluating the revised manuscript and the rebutta, I find that the authors have adequately responded to the major issues I raised earlier.

---

> > > ### Author Response · Authors · 2026-04-03
> > >
> > > We appreciate your response and are glad our responses addressed your concerns. Thanks again for your careful consideration of our work!

---

### Decision · Program_Chairs · 2026-04-30

**Decision:**

Accept (regular)

**Comment:**

The paper proposes LCSA, a likelihood-constrained adversarial attack framework that leverages structural causal models and a ripple perturbation mechanism to generate structurally consistent tabular adversarial examples.

The method is technically sound and well-motivated with clear gains in both attack success and structural validity across diverse settings.
The rebuttal effectively addressed most concerns, and several reviewers updated to fully resolved.

There are some remaining issues such as computational overhead, reliance on DAG assumptions, and limited reproducibility (code not yet released. I hope authors can address them in revision.

Overall, the work offers a meaningful shift toward structure-aware attacks in tabular domains and is likely to inspire follow-up research.